# First bone-cracking dog coprolites provide new insight into bone consumption in *Borophagus* and their unique ecological niche

Xiaoming Wang[1,2,3]*, Stuart C White[4], Mairin Balisi[1,3], Jacob Biewer[5,6], Julia Sankey[6], Dennis Garber[1], Z Jack Tseng[1,2,7]

[1]Department of Vertebrate Paleontology, Natural History Museum of Los Angeles County, Los Angeles, United States; [2]Department of Vertebrate Paleontology, American Museum of Natural History, New York, United States; [3]Department of Ecology and Evolutionary Biology, University of California, Los Angeles, United States; [4]School of Dentistry, University of California, Los Angeles, United States; [5]Department of Geological Sciences, California State University, Fullerton, United States; [6]Department of Geology, California State University Stanislaus, Turlock, United States; [7]Department of Pathology and Anatomical Sciences, Jacobs School of Medicine and Biomedical Sciences, University at Buffalo, Buffalo, United States

**Abstract** Borophagine canids have long been hypothesized to be North American ecological 'avatars' of living hyenas in Africa and Asia, but direct fossil evidence of hyena-like bone consumption is hitherto unknown. We report rare coprolites (fossilized feces) of *Borophagus parvus* from the late Miocene of California and, for the first time, describe unambiguous evidence that these predatory canids ingested large amounts of bone. Surface morphology, micro-CT analyses, and contextual information reveal (1) droppings in concentrations signifying scent-marking behavior, similar to latrines used by living social carnivorans; (2) routine consumption of skeletons; (3) undissolved bones inside coprolites indicating gastrointestinal similarity to modern striped and brown hyenas; (4) *B. parvus* body weight of ~24 kg, reaching sizes of obligatory large-prey hunters; and (5) prey size ranging ~35–100 kg. This combination of traits suggests that bone-crushing *Borophagus* potentially hunted in collaborative social groups and occupied a niche no longer present in North American ecosystems.
DOI: https://doi.org/10.7554/eLife.34773.001

*For correspondence:
xwang@nhm.org

Competing interests: The authors declare that no competing interests exist.

## Introduction

Several lineages of dogs (family Canidae) and hyenas (family Hyaenidae) have independently evolved striking bone-crushing adaptations, such as highly robust skulls, jaws, teeth, and large attachment areas for powerful masticatory muscles. These highly specialized bone-cracking morphologies are likely associated with social hunting (*Van Valkenburgh and Koepfli, 1993*; *Van Valkenburgh et al., 2003*), as best exemplified by the living spotted hyena (*Kruuk, 1972*). Spotted hyenas hunt and feed in groups, have a gastrointestinal system that is able to break down large quantities of bone consumed, and discharge feces with high-carbonate content (*Macdonald, 1978*; *Estes, 1991*; *Hulsman et al., 2010*). Whether extinct hyena-like canids consumed a comparable quantity of bone—and, if so, how those bones are processed inside their gastrointestinal system—both remain questions that have been unanswerable for lack of fossil evidence. These questions are directly addressed in this study with the first discovery of coprolites (fossilized feces) from one of the

**eLife digest** Living hyenas are infamous for crushing the bones of their prey to extract the nutritious marrow inside. This feeding ability is rare today, and African and Asian hyenas, particularly the spotted hyena, are the only true 'bone-crackers' in our modern ecosystems. Yet, between 16 to 2 million years ago, the common, but now extinct North American dogs also crushed bone. Their skeletal features – such as highly robust skulls and jaws, teeth to withstand high stress, and large muscle-attachment areas for a powerful bite –share many similarities with the spotted hyena. It is therefore likely that these extinct North American dogs played a similar role in the ecosystem as living hyenas do now.

The last of these bone-cracking dogs, *Borophagus,* vanished approximately 2 million years ago. In a recent study in 2018, researchers discovered fossilized feces, also known as coprolites, which presumably belong to *Borophagus parvus* that lived in central California between 5 to 6 million years. These coprolites preserve ingested bone and so provide more evidence of what this species of dogs ate. Now, Wang et al. – including some of the researchers involved in the previous study – analyzed the fossil coprolites and their ingredients in great detail using computer tomography, measurements and comparisons with living predators and their prey.

The results show that *Borophagus parvus* weighed around 24 kg and hunted large prey of 35 kg up to 100 kg: the size of a living mule deer. Its skull structure was similar to the spotted hyena, but its digestive system resembled that of striped and brown hyenas. Spotted hyenas have chalk white feces containing digested bone matter, presumably due to a highly acidic digestive system, but the coprolites of *Borophagus* contained undissolved bones (which they ate regularly). Wang et al. also discovered that these dogs dropped feces in clusters, which is how the spotted hyena and wolves mark territory. This suggests that *Borophagus* were also social animals.

Bone-crackers (modern and extinct) act as apex predators and providers of free organic material needed for decomposition, which are essential roles for maintaining a healthy ecosystem. The extinction of *Borophagus* likely modified the dynamics of the food web over the past few million years. It remains unclear why this way of feeding is absent in all living animals of North America. Future studies could investigate how the disappearance of *Borophagus* may have influenced the establishment of modern environments, eventually setting the scene for human habitation of the continent.

DOI: https://doi.org/10.7554/eLife.34773.002

archetypal bone-eating dogs, *Borophagus.* Borophagines are a group of carnivorans with highly specialized craniodental morphological traits indicative of bone-cracking adaptation, and have long been recognized to be a terminal member of the subfamily Borophaginae that went extinct just before the beginning of the Ice Ages in North America (*Wang et al., 1999*). Therefore, understanding the paleoecology of these top predators has important implications for reconstructing community dynamics on the continent before megafaunal extinction and human habitation.

We analyzed a new sample of coprolites recently discovered from the Mehrten Formation (latest Miocene, 5.3–6.4 Ma) in Stanislaus County, California. Numerous bone fragments on the external surface and inside the coprolites strongly suggest that they were produced by *Borophagus*, which is amply represented by body fossils at the same fossil-producing area, thereby affording a rare opportunity to directly examine the diet of an extinct bone-crushing top predator. Despite improvements in our understanding of the biomechanics of the functional convergence of craniodental adaptations between Eurasian-African hyaenids and North American borophagine canids (*Werdelin, 1989*; *Tseng and Wang, 2010*; *Tseng and Wang, 2011*), dietary inferences were previously made only from the fossilized bones of these predators. The discovery of coprolites thus offers the first glimpse into the food ingested and excreted by these 'hyaenoid dogs' (*Simpson, 1945*), as well as several traits related to their territorial behavior, social hunting, and bone digestion that were previously unapproachable. This study examines 14 coprolites recovered from two localities in the Turlock Lake area, as well as their presumed producer, *Borophagus parvus*. Our findings provide new insights into the paleoecology of this group of top predators and refine their position in the food web at the end of the Miocene Epoch in North America.

### Institutional abbreviations

F:AM Frick Collection of the American Museum of Natural History, New York, New York; FMNH, Field Museum of Natural History, Chicago, Illinois; LACM, Natural History Museum of Los Angeles County, Los Angeles, California; UCMP, Museum of Paleontology at University of California, Berkeley, California.

## Results and discussion

### Producer of Mehrten coprolites

The large number of bones inside most Mehrten coprolites rules out herbivores as their producers. The size of the coprolites further indicates large carnivorans as their original makers. For medium to large carnivorans from the Mehrten Formation, *Wagner (1976, 1981)* listed a bone-crushing dog *Borophagus secundus* (=*Osteoborus cyonoides*), a small coyote-sized *Eucyon davisi*, an ancestral badger *Pliotaxidea garberi*, an early wolverine *Plesiogulo marshalli*, and an ancestral cat *Pseudaelurus* near *P. hibbardi*. Most recently, *Balisi et al. (2018)* added a fox, *Vulpes stenognathus*, to the list. Of the above, *Vulpes*, *Eucyon*, *Pliotaxidea*, and *Plesiogulo* can be ruled out as being too small to produce scats of the size of the Mehrten coprolites, whereas the true nature of Mehrten felids is poorly known.

Of the large Mehrten canids, *Balisi et al. (2018)* recognized two bone-crushing canids, *B. secundus* and *B. parvus*, which are the only wolf-sized taxa large enough to be the producers of the Mehrten coprolites. Of these two species, *B. secundus* is rare, represented by two fragmentary jaws and teeth plus 1–2 questionably referred teeth, whereas *B. parvus* is far better represented by 27 specimens. At the main coprolite-producing locality (see Materials and methods), LACM locality 3937 (=Dennis Garber T-34 locality), an isolated P2 or P3 (UCMP 235515) is questionably referred to *B. secundus* (*Balisi et al., 2018*), whereas in LACM locality 3935, no identifiable carnivoran is found (Figure 9).

The Mehrten coprolites are comparable in size (Table 1) to scats from extant wolves and are generally larger than those from living coyotes, despite significant overlap between scat diameters of the wolves (average 27 mm, range 13–47 mm) and the coyotes (average 21 mm, range 7–34 mm) (*Weaver and Fritts, 1979*; *Reid, 2015*). In extant African carnivores, *Harrison (2011)* documented scat diameters of 20–35 mm from African hunting dogs, *Lycaon pictus*, and striped hyena, *Hyaena hyaena*. Therefore, with a maximum diameter of 31.2 mm, the Mehrten coprolites are more likely produced by a wolf-sized *Borophagus* than a coyote- to fox-sized *Eucyon*. Of the two species of Mehrten *Borophagus*, *B. parvus* was the more likely producers of Mehrten coprolites based on their body size and far better representation of body fossils, although the possibility of *B. secundus* cannot be excluded.

### Coprolite morphology

We adopt a modified scheme for characterizing hyaenid coprolite aggregate pellets introduced by *Diedrich (2012)*, but we use different terminologies for orientations (*Figure 1A*). Although scat morphology of extant wolves and hyenas may be somewhat different—depending on length of retention in digestive tract, fiber and water content of feces, and hardness of ground on which scats were dropped—our Mehrten coprolites (*Figures 2* and *3*) appear to share substantial similarities to those of living hyenas (*Figure 1B*). Of the 14 individually catalogued coprolites, five probably are a first dropping due to their bluntly constricted terminal on at least one of their ends and their relatively greater diameter (LACM 158707, 158708, 158709, 158711, and 158712). However, only one, LACM 158709, has the typical shape of a conical pellet (*Figure 1A*), although LACM 158707 represents a variation of the conical-disk pellet combination that failed to separate after dropping. LACM 158708 has tapering on both ends, suggesting that the modern hyena pellet terminology by *Diedrich (2012)* does not completely apply to the Mehrten canids. The rest of the nine pieces are all incomplete pellets, and their exact position within the scat string is difficult to determine.

If the above assessment is correct, the Mehrten coprolite sample probably consists of individual pellets from multiple dropping events possibly by multiple individuals. This is also suggested by different degrees of desiccation among different coprolite pellets (*Figure 5E*), that is, they were not defecated at the same time. If this is the case, and assuming that the coprolites have not been

transported (there is no sign for transportation), the LACM 3937 locality may have been an ancient 'latrine' ground for social defecating and scent-marking for territorial boundaries. Such locations have been well documented in extant spotted hyenas (*Kruuk, 1972*), coyotes (*Gese and Ruff, 1997*), and wolves (*Asa et al., 1985*; *Harrington and Asa, 2003*). While such behavior is common among social carnivorans, it has not been documented in extinct carnivorans.

Mehrten coprolites maintain nearly perfectly rounded cross sections, showing no sign of post-defecation settling or flattening, nor is there any sign of deformation during the initial impact of dropping. This suggests that the original feces were able to maintain their integrity either because of a relatively hard, moisture-free matrix, and/or because the bones inside plus the high-calcareous contents of the matrix resulted in relatively rigid feces at defecation. Nor do the coprolites show major signs of post-defecation alteration, suggesting fast burial after dropping. Bones are abundant in all coprolites, consisting of 5% of total volume of all coprolites (range 2–25%; see *Table 1* for individual volume estimates). As examples, we describe two complete coprolites below.

## LACM 158707 (*Figure 2*, *Video 1*)
This is a nearly perfectly preserved coprolite and also one of the largest, measuring 31.2 mm in maximum diameter. The bluntly tapered end suggests a terminal pellet (first dropping, *Figure 1A*). This coprolite is composed of two unseparated pellets, as delineated by a visible groove. The proximal (last dropping) end has a flat surface, representing a clean separation from the next pellet. A single bone is visible on the external surface, with at least 14 bone fragments recognizable in CT image (*Figure 2D*), although all are unidentifiable small pieces.

## LACM 158708 (*Figure 3*, *Video 2* )
This is another of the most complete coprolite pellets. The cross-section is nearly perfectly rounded, although there is a distinct flattening on one side, indicating dropping on hard ground during defecation. This coprolite also contains the largest piece of bone, a fragment of a rib shaft measuring 29 mm long × 9.2 mm wide × 5.1 mm thick, that nearly spans the length of the pellet (*Figure 3C,D*). The terminal end of this rib also protrudes outside the coprolite on the tapered end, leaving a sharp tip, 3 mm long, projecting at an angle into the lateral wall of the intestine and showing modest polishing on its surface (red arrows in *Figure 3*). Another piece of bone (enclosed by red dashed line in *Figure 3*) has a rounded external surface with a thin cortex filled entirely by cancellous bone, suggesting an articular joint. The size of this bone is consistent with a rib head for the shaft, although we cannot positively identify this as such without physical preparation. Two other smaller pieces of bones are also identified from microCT-scanned images. Total bone volume is 13% of coprolite matrix for LACM 158708, among the highest of all coprolites (*Table 1*).

## Bones inside coprolites
The majority of bones inside the coprolites, even when fully exposed, are too small and too fragmentary to be identified to a particular element or to a particular taxon beyond mammals or even vertebrates. Such difficulty can also be compounded by digitally segmented microCT reconstructions. These digitally separated bones are often an inexact replication of the actual shapes, mostly due to high similarity in X-ray opacity between bones and surrounding matrix. With the exception of a single rib fragment in LACM 158708, all other virtually segmented bones lack sufficient morphological detail to be unambiguously identified.

Generally, there is a lack of clear orientation relative to the long axis of each coprolite (*Figure 4*). This randomness may be a result of several factors. With the exception of the rib fragment—which, because of its length, must be aligned along the long axis of the coprolite (*Figure 3C,D*)—most bones are relatively small, and intestine diameter is not a limiting factor in their orientation. Lack of a longitudinal orientation may also be due to a relatively viscous (low water content) matrix and compaction during the last (dehydration) journey of feces through the large intestine.

Surface modifications on bones include rounding of corners, polishing of surface, and acid etching. The external surface of a small bone (red dashed line in *Figure 3B*) exposed to the intestine wall has experienced visible polishing; this polished surface was also stained a darker color than the unpolished parts. Polishing is known to occur in 80% of bones in extant wolf scat ([*Esteban-Nadal et al., 2010*]:Figure 18). Etching and flaking are seen on an exposed bone in LACM 158707

(*Figure 2D,E*); this is relatively uncommon in the scat of extant wolves, occurring in only 0.9% of bones contained in wolf scat ([*Esteban-Nadal et al., 2010*]:Figure 18).

About 5% of bones recovered from living wolf scat can be identified to their prey species (*Fosse et al., 2012*). Four bone fragments from Mehrten coprolites, consisting of 8% of the total number of bone fragments (*Table 1*), preserve enough original morphology to be narrowed to more specific taxa or anatomic structure. They are described below.

### Bird limb bone in LACM 158711 (c2 in *Figure 5C*)

A large piece of bone, measuring at least 6.0 × 3.6 mm in cross-sectional area, has thin walls and hollow internal structure likely belonging to a bird limb bone. The thickness of the wall is 0.8 mm, and it has many thin struts on its internal surfaces. The extreme hollowness of this bone is in sharp contrast to an adjacent limb bone fragment (c1 in *Figure 5C*) that has a cortical thickness of 4.6 mm. The only fossil bird so far reported from the Modesto Reservoir Member is a goose, *Branta* (*Wagner, 1981*).

### Ascending ramus of the dentary of a beaver in LACM 158712 (d1 in *Figure 5D*)

A very large flat bone, resembling the ascending ramus of a beaver, spans the width of the coprolite. The anterior and dorsal rims are intact. The posterior rim is very thin-walled, and we cannot be certain if its true border is completely intact. Part of the latter is smooth enough to be possibly intact; if so, we tentatively identify this bone as a beaver ascending ramus. Two beavers, *Castor californicus* and *Dipoides vallicula*, are both known in the Mehrten Formation (*Wagner, 1981*) and belong to the semiaquatic beaver clade (*Rybczynski, 2007*). All of them possess a highly diagnostic backwardly hooked ascending ramus (coronoid process; see comparison with that of *Eucastor tortus* in *Figure 5G*, an extant species adjacent to *Dipoides* in the beaver phylogeny) that is possibly related to woodcutting behavior. However, because of the poorly preserved posterior border, we cannot rule out the possibility of this bone being the distal portion of a scapula. (If a scapula, it probably belongs to a medium dog-sized mammal).

### Basicranium of a medium-sized mammal in LACM 157716 (f1 in *Figure 5F* and digital reconstruction in *Figure 5H*)

An incomplete, deep foramen measuring 3.0 mm in maximum length and 2.1 mm in minimal width is rimmed by an incomplete shelf on one side and a beam-like structure on the other. There is a small nutrient foramen, less than 1 mm across, on the wall of the broader side of the larger foramen. Such a configuration is most frequently seen in the basicranial region, such as the foramen ovale anterior to the tympanic bulla in living *Odocoileus*, which also has a nutrient foramen on the medial wall of the foramen ovale. However, the size of this foramen on LACM 158716 and its detailed anatomy do not match exactly with *Odocoileus*. Given the poor preservation, this bone is not easily identified beyond its anatomic position.

### Rib of a large mammal in LACM 158708 (*Figure 3*)

A gentle curvature and elongated shape with four relatively straight walls make this bone easily identified as the proximal segment of a rib. Modern wolves typically feed on the internal organs first—such as the heart, lungs, and liver (*Stahler et al., 2006*)—and the ribcage is the main obstacle to reaching the organs. Experimental data also suggest that mammalian ribs are a highly desirable part of modern wolf diet, with about 99% of ribs being consumed (i.e. 1% left uneaten) and the ribs being the most frequently (25%) identified bone fragments recovered from scat ([*Klippel et al., 1987*]:Tables 1 and *2*). The large size of this bone (>30 mm in length) is unusual, typically comprising a small percentage of carnivore scats ([*Kolska Horwitz, 1990*]:*Table 1*). With the large size of this bone, we can estimate the minimum body size range of prey (see Figures 10–12).

## Comparison to modern wolf scats

Experimental data on modern gray wolf diet and their scat permit a certain measure of quantifying scat contents and identifying prey items. However, most of these methods are based on sorting soft matter in wolf feces (e.g. [*Floyd et al., 1978*; *Weaver, 1993*]), which are typically not preserved in

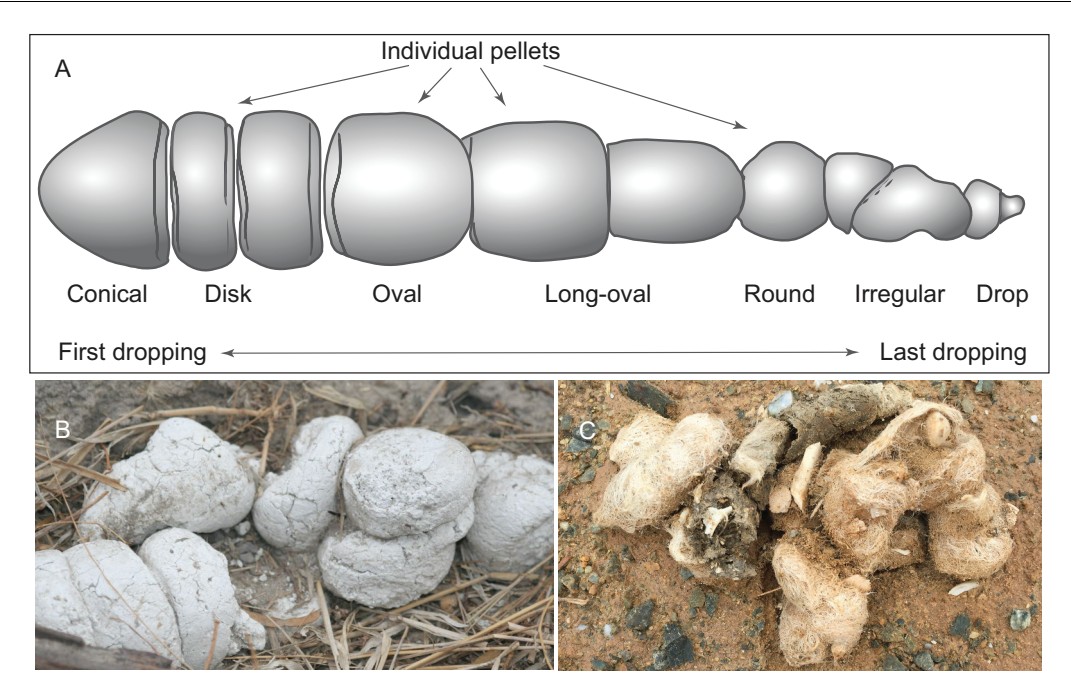

**Figure 1.** (A) morphology of individual pellets in a complete assemblage of feces from a single dropping event by the extant spotted hyena, Crocuta crocuta. Adapted from ([*Diedrich, 2012*]:*Figure 4*), except for the orientation (Diedrich's anterior/posterior orientation is counter to traditional sense of anatomy). (B) scats of extant spotted hyena (still image reproduced from a Smithsonian magazine video, available at http://www.smithsonianmag.com/videos/category/weird-science/weird-science-hyena-poop/?no128ist). (C) scats of extant grey wolf; note preservation of bone fragments and hairs (photo by Xiaoming Wang on September 21, 2016 in Xorkol Basin in southern Xinjiang Uygur Autonomous Region, China).
DOI: https://doi.org/10.7554/eLife.34773.003

coprolites. A study on bone fragments preserved in extant Iberian wolf (*Canis lupus signatus*) scat provides a valuable basis for comparison (*Esteban-Nadal et al., 2010*). In the Spanish samples, the numbers of skeletal fragments per scat vary from one to 96 ([*Esteban-Nadal et al., 2010*]:*Table 2*), the upper limit being substantially more than those in our fossil samples. These higher numbers of bones can probably be explained by two factors. First, although Esteban-Nadal et al. did not specifically state it, their count of a scat almost certainly includes the entire ejected feces in a single dropping event, in contrast to our own treatment of a single individual piece of coprolite. (our disarticulated pieces of coprolites correspond to individual pellets of a long series of scat described in [*Diedrich, 2012*]:*Figure 4*) (*Figure 1*). Second, bones from extant wolf scat are exhaustively sampled (picked through dry samples and/or screened after chemical treatments) in contrast to our visual inspections in microCT-scanned images. Small bones that have similar radio-opacity as the surrounding matrix can potentially be missed in the counts (*Table 1*). If we discount the above two factors, fossil coprolites from the Mehrten possibly contain numbers of bone fragments comparable to those in extant Iberian wolf scat.

More than 80% of bone fragments in Iberian wolf scat are not identifiable to a particular bone or taxon. A study of Polish wolves had a 95% rate of unidentified bones (*Fosse et al., 2012*). The same is true for Mehrten canid coprolites: four relatively large bones are identified among 48 in total (i.e. 92% unidentified bones). Finally, sizes of individual bone fragments in scat of extant wolves are also roughly comparable to those in our fossils. The digested bones have a rather uniform size range of 1–2 cm to a few mm in diameter.

## Bone crushing adaptations in *Borophagus*

*Borophagus* and bone-cracking hyaenids such as *Crocuta* share several craniodental features that have been interpreted as adaptations for a durophagous diet. These include robust cheek teeth often exhibiting heavy cusp wear (the lower p4 and m1 in *Borophagus* and lower p3 and p4 in

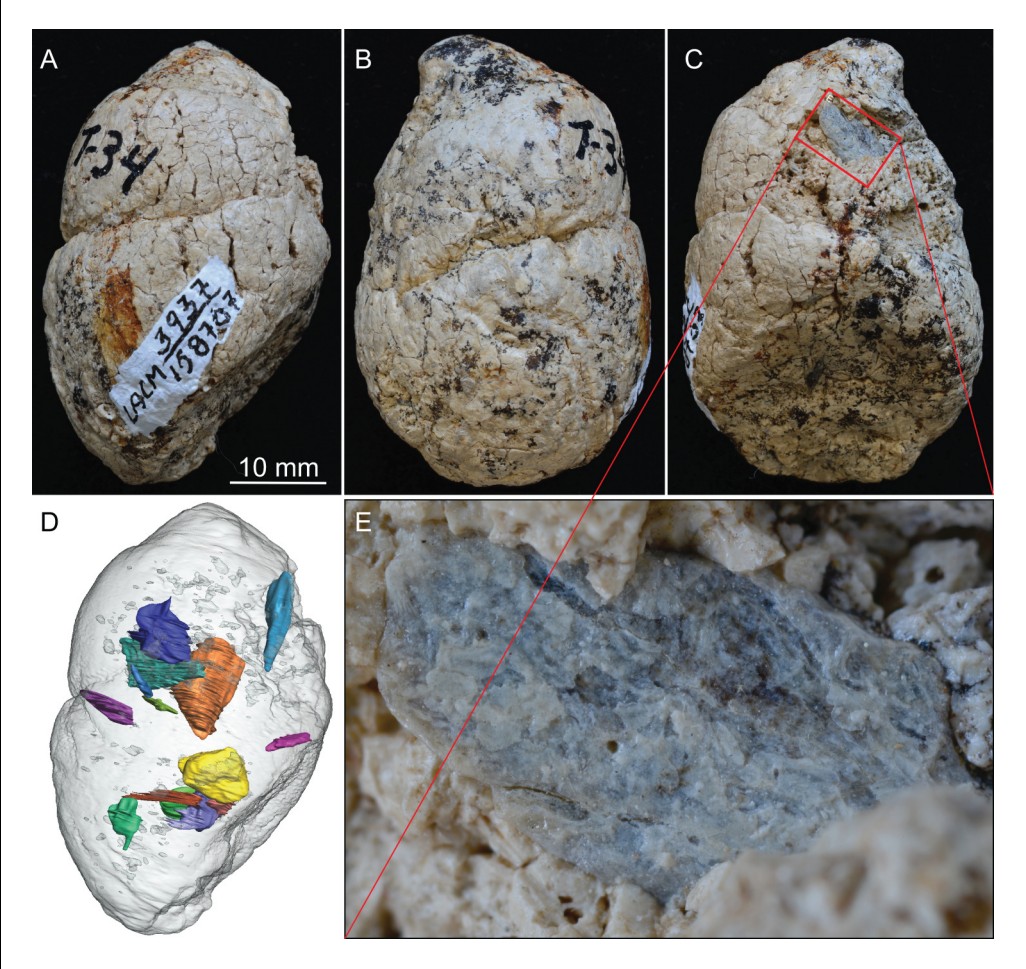

**Figure 2.** LACM 158707, a complete coprolite from LACM locality 3937 (=Turlock Lake 34), Mehrten Formation, Stanislaus County, California, collected by Dennis Garber. (**A**) Lateral view, top is toward distal (first dropping) end; (**B**) another lateral view about 90° from A; (**C**) another lateral view about 90° of further rotation from B; (**D**) 14 bone fragments (in various colors) digitally segmented within the coprolite (light grey) in the same orientation as in A; (**E**) close-up of an exposed bone fragment (unidentified) on C showing acid etching (flaking) on external surface. See also *Video 1* to show three dimensional relationships of individual bones within this coprolite.
DOI: https://doi.org/10.7554/eLife.34773.004

*Crocuta*; *Figure 6A*). Upper and lower dentitions of both taxa also exhibit specialized enamel microstructure (Hunter-Schreger Bands) of the cheek teeth interpreted to represent evolutionary responses to resisting increasingly hard and abrasive foods (*Rensberger and Wang, 2005*; *Tseng, 2011*). In both dental morphology and enamel microstructure, *Borophagus* and *Crocuta* share more similarities to each other than to *Canis* (*Figure 6A,C*). However, macrowear analyses of the lower carnassial tooth (m1) in population samples of the three carnivorans demonstrate that the extant bone-cracking *Crocuta* exhibits much more extreme cusp wear on average than either *Canis* or *Borophagus* (*DeSantis et al., 2017*) (*Figure 6B*). In terms of cranial shape, *Crocuta* is intermediate between *Borophagus* and *Canis* in having a moderately elongate rostrum and a moderately smooth forehead, whereas *Borophagus* has the combination of a relatively short rostrum with a more 'stepped' appearance of the forehead (*Figure 6D–E*). Nevertheless, within the phylogenetic context of their respective lineages, *Borophagus* and *Crocuta* represent similar extremes along an evolutionary morphological continuum, with *Canis* located beyond the morphospace occupied by either borophagine canids or hyaenids (*Tseng and Wang, 2011*; *Balisi et al., 2018*) (*Figure 6E*). Lastly, comparisons of overall stress distributions during unilateral carnassial (P4) bite simulations using finite element analysis indicate that the crania of *Crocuta* and *Borophagus* are more similar to each

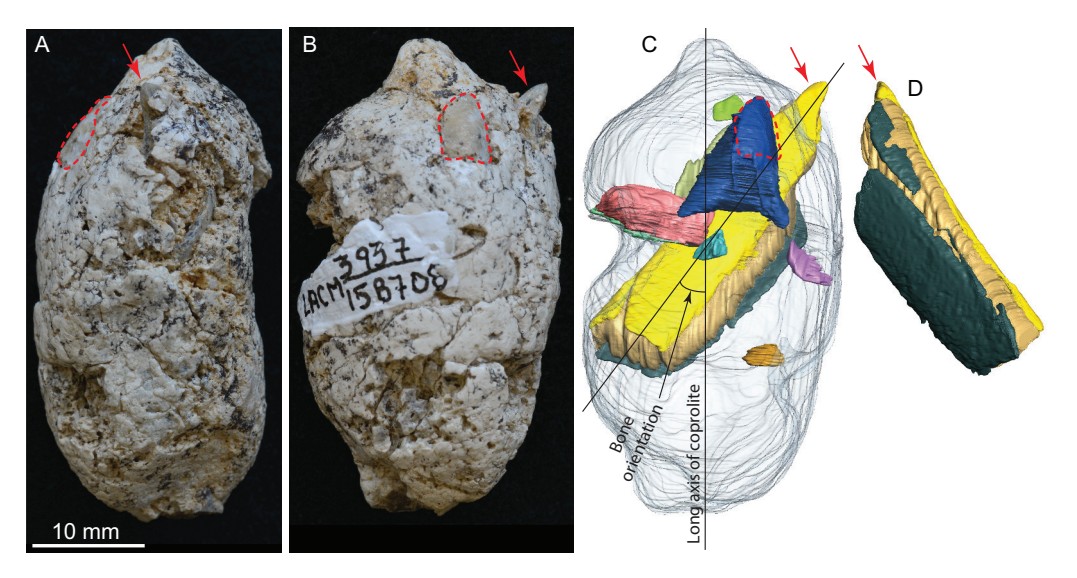

**Figure 3.** LACM 158708, a complete coprolite and bones contained within, from LACM locality 3937, Mehrten Formation, Stanislaus County, California, collected by Dennis Garber. (**A**) Lateral view, top is toward distal (first dropping) end; (**B**) another lateral view about 90° from A; (**C**) digitally separated individual bones (in different colors) within coprolite matrix (light grey), identical view as that of B; (**D**) a rotated view of a rib fragment seen in C, showing the convex (external) side, yellow and dark green shapes representing internal (toward chest cavity) and external cortical bone respectively, and yellowish brown sandwiched between the cortical bones being cancellous bone. Red arrows indicate the same protruded tip of rib fragment, and red dashed lines define the exposed outlines of a flat bone (mostly buried within coprolite matrix; dark blue piece in C shows the full extent of this bone within the coprolite). With the exception of the rib, all other bone fragments are unidentifiable. See also *Video 2* and original Avizo segmentation file (**web link**) to show three dimensional relationships of individual bones within this coprolite.
DOI: https://doi.org/10.7554/eLife.34773.005

other in exhibiting lower and more dissipated stress patterns than *Canis* (*Tseng, 2011*) (*Figure 6F*). These functional morphological characteristics (except for the macrowear data of *Borophagus*, newly presented here) have been used to justify classifications of both *Borophagus* and *Crocuta* as specialized bone-cracking ecomorphs.

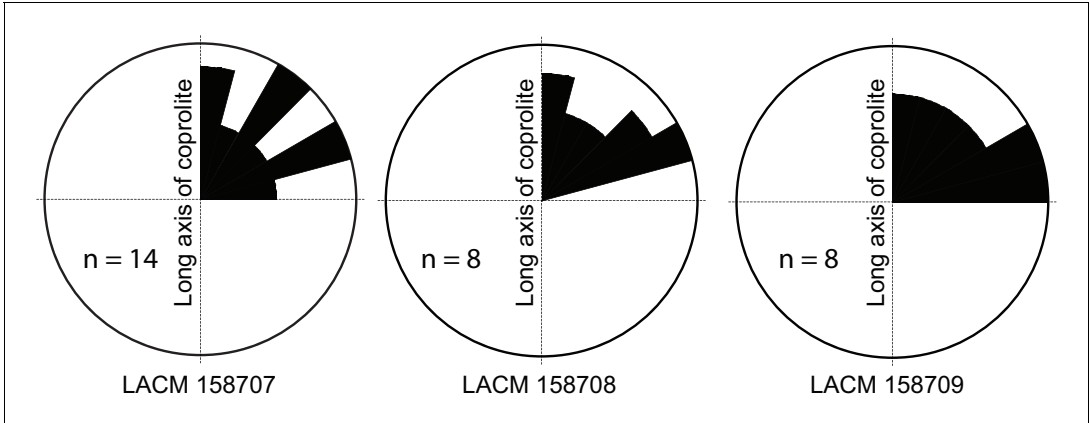

**Figure 4.** Rose diagram of bone orientations inside the coprolites. Only coprolite pellets with at least eight bone fragments inside and a clear long axis are presented. (In the case of LACM 158709, see *Figure 3A*: although its axial dimension is similar to its diameter, its constricted distal end gives unambiguous orientation of its long axis.) Angles (0–90°) are between the long axis of the coprolite and the long axis of bone fragments in three-dimensional space (see *Figure 3C* for a definition of the angles). Data from *Table 1*.
DOI: https://doi.org/10.7554/eLife.34773.006

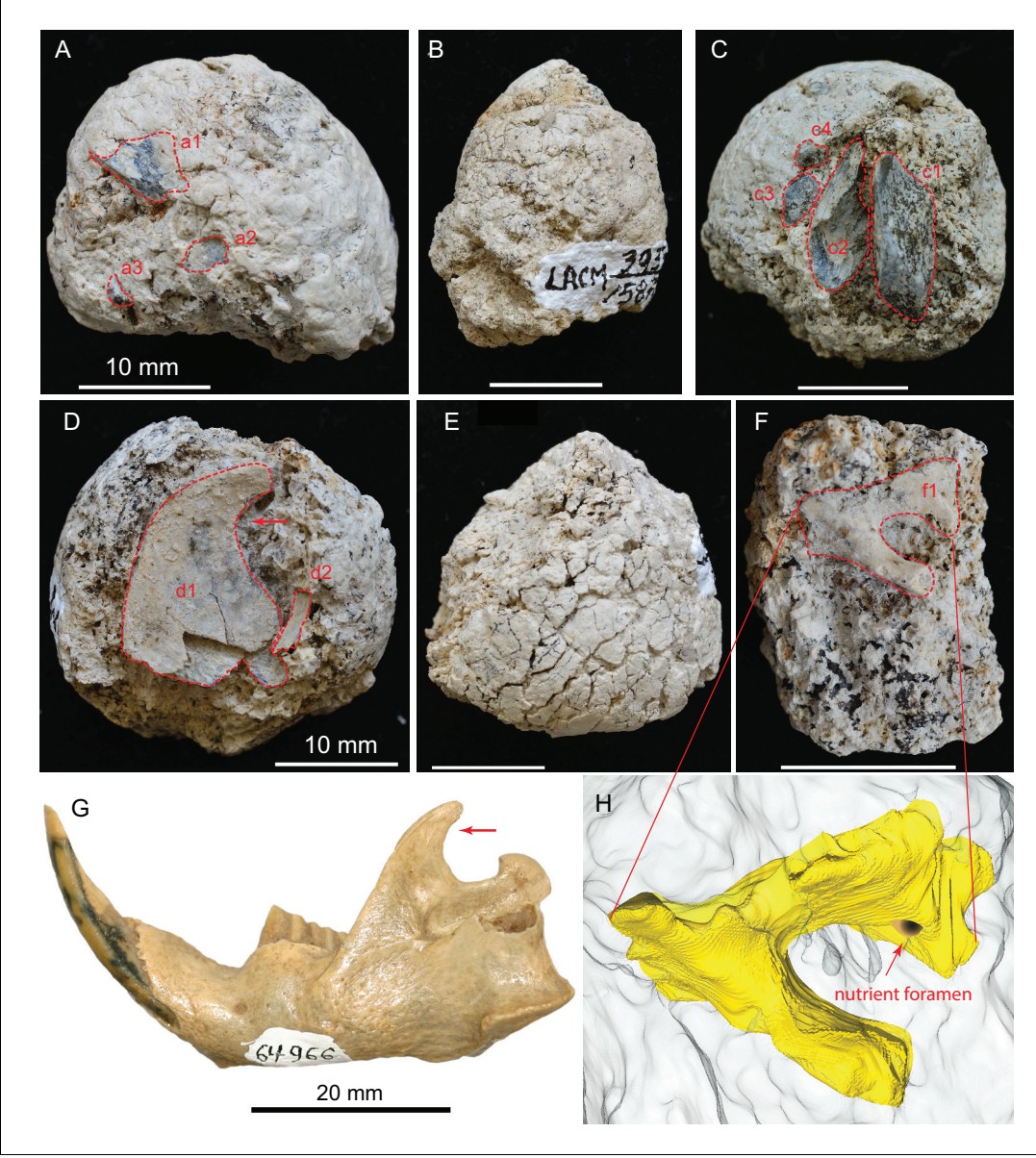

**Figure 5.** (**A**) LACM 158709 with three visible bone fragments (a1, a2, a3); (**B**) LACM 158710; (**C**) LACM 158711 with four visible bone fragments partially prepared (c1–c4); (**D**) LACM 158712 with two visible bone fragments partially exposed (d1, d2); (**E**) LACM 158713, surface cracks suggesting desiccation before burial; (**F**) LACM 158716 with one bone fragment partially exposed (f1); (**G**) left jaw of extant *Eucastor tortus*, compared to the fragment of coronoid process of the mandible (red arrows) of d1 in D (FMNH 64966; photo courtesy of Joshua Samuels); (**H**) digitally reconstructed bone (colored yellow; light grey background is coprolite matrix) of f1 in F, tentatively identified as the ventral aspect of the foramen ovale in the basisphenoid of a medium-sized mammal. Dashed red lines indicate exposed outlines of bones. All scales for coprolites are 10 mm.

DOI: https://doi.org/10.7554/eLife.34773.007

## Reconstructing the potential ecological role of *Borophagus*

### Bone consumption

Bones as a supplementary food source must be a net gain if the cost of processing bones (both ingestion and digestion) can be managed. The average compact bone consists of about 30% organic matrix (*Eastoe and Eastoe, 1954*; *Guyton and Hall, 2006*), mostly collagen fibers. The organic component (nutritional value) is even higher if marrow in the cancellous bone is also considered. Living spotted hyenas in Africa possess morphological and physiological adaptations that permit efficient

**Table 1.** Measurements of coprolites and their included bones.

Maximum diameter and length of coprolites are measured by digital calipers, and the rest are calculated by Avizo software. '*' in coprolite diameter and length indicates incomplete dimensions due to damage.

| LACM catalogue number | Maximum diameter × length (mm) | Coprolite volume (mm³) | Matrix volume (mm³) | Matrix fraction | Bone fragments contained | Bone max length (mm) | Bone max width (mm) | Bone orientation (degrees) | Bone volume (mm³) | Bone fraction/ coprolite |
|---|---|---|---|---|---|---|---|---|---|---|
| | | Coprolite dimensions | | | | | Bone dimensions | | | |
| 158706 | 24.6* × 31.9* | 5871 | 5871 | 100% | None | | | | | |
| 158707 | 31.2 × 47.2 | 18508 | 17823 | 96% | Bone_1 | 8.1 | 6.6 | 70 | 155 | |
| | | | | | Bone_2 | 16.5 | 5.1 | 24 | 56 | |
| | | | | | Bone_3 | 10.6 | 7.8 | 8 | 207 | |
| | | | | | Bone_4 | 6.1 | 5.1 | 70 | 23 | |
| | | | | | Bone_5 | 4.6 | 4.6 | 63 | 26 | |
| | | | | | Bone_6 | 7.9 | 7.2 | 10 | 43 | |
| | | | | | Bone_7 | 3.6 | 2.4 | 32 | 11 | |
| | | | | | Bone_8 | 11.4 | 4.7 | 12 | 45 | |
| | | | | | Bone_9 | 5.5 | 1.7 | 85 | 3 | |
| | | | | | Bone_10 | 4.7 | 2.8 | 40 | 5 | |
| | | | | | Bone_11 | 11.0 | 2.2 | 68 | 8 | |
| | | | | | Bone_12 | 8.5 | 4.8 | 43 | 78 | |
| | | | | | Bone_13 | 5.0 | 3.9 | 58 | 10 | |
| | | | | | Bone_14 | 6.1 | 3.6 | 37 | 14 | |
| | | | | | Total bone | | | | 685 | 4% |
| 158708 | 24.9 × 44.6 | 10184 | 8814 | 87% | Bone_1 | 3.0 | 2.1 | 21 | 3 | |
| | | | | | Bone_2 | 11.1 | 4.4 | 15 | 25 | |
| | | | | | Bone_3 | 16.1 | 10.0 | 14 | 344 | |
| | | | | | Bone_4 | 5.6 | 3.4 | 39 | 11 | |
| | | | | | Bone_5 Cortex | 8.3 | 7.7 | 61 | 14 | |
| | | | | | Bone_5 Marrow | 9.7 | 6.8 | 61 | 104 | |
| | | | | | Bone_6 | 5.1 | 3.4 | 49 | 11 | |
| | | | | | Bone_7 | 4.3 | 2.1 | 70 | 6 | |
| | | | | | Bone_8 Rib long | 31.0 | 7.0 | 46 | 156 | |
| | | | | | Bone_8 Rib Marrow | 30.1 | 8.2 | 46 | 574 | |
| | | | | | Bone_8 Rib short | 29.4 | 8.7 | 46 | 122 | |
| | | | | | Total bone | | | | 1370 | 13% |
| 158709 | 27.1 × 23.4 | 6556 | 6341 | 97% | Bone_1 | 12.5 | 2.3 | 65 | 21 | |
| | | | | | Bone_2 | 7.9 | 4.5 | 80 | 45 | |
| | | | | | Bone_3 | 12.6 | 4.4 | 24 | 53 | |
| | | | | | Bone_4 | 6.6 | 2.3 | 45 | 10 | |
| | | | | | Bone_5 | 5.0 | 2.2 | 54 | 10 | |
| | | | | | Bone_6 | 3.6 | 1.9 | 72 | 5 | |
| | | | | | Bone_7 | 6.1 | 4.5 | 11 | 28 | |
| | | | | | Bone_8 | 10.6 | 4.9 | 81 | 42 | |

*Table 1 continued on next page*

Table 1 continued

| LACM catalogue number | Maximum diameter × length (mm) | Coprolite volume (mm³) | Matrix volume (mm³) | Matrix fraction | Bone fragments contained | Bone max length (mm) | Bone max width (mm) | Bone orientation (degrees) | Bone volume (mm³) | Bone fraction/ coprolite |
|---|---|---|---|---|---|---|---|---|---|---|
| | | | | | Total bone | | | | 214 | 3% |
| 158710 | 21.3 × 26.6 | 4066 | 4066 | 100% | None | | | | | |
| 158711 | 29.1* × 31.2* | 11741 | 11251 | 96% | Bone_1 | 16.9 | 7.1 | | 197 | |
| | | | | | Bone_2 | 18.7 | 6.3 | | 93 | |
| | | | | | Bone_3 | 6.8 | 3.0 | | 13 | |
| | | | | | Bone_4 | 13.1 | 7.1 | | 169 | |
| | | | | | Bone_5 | 6.5 | 3.7 | | 18 | |
| | | | | | Total bone | | | | 490 | 4% |
| 158712 | 29.4 × 27.5* | 8284 | 8012 | 97% | Bone_1 | 21.3 | 12.3 | 37 | 234 | |
| | | | | | Bone_2 | 13.2 | 3.1 | 67 | 18 | |
| | | | | | Bone_3 | 7.1 | 2.5 | 2 | 7 | |
| | | | | | Bone_4 | 3.4 | 1.9 | 18 | 2 | |
| | | | | | Bone_5 | 8.6 | 2.8 | 75 | 12 | |
| | | | | | Total | | | | 272 | 3% |
| 158713 | 27.5 × 25.6* | 8694 | 8454 | 97% | Bone_1 | 15.2 | 6.5 | 26 | 107 | |
| | | | | | Bone_2 | 10.4 | 8.7 | 44 | 114 | |
| | | | | | Bone_3 | 4.7 | 1.9 | 66 | 5 | |
| | | | | | Bone_4 | 4.9 | 2.7 | 72 | 14 | |
| | | | | | Total bone | | | | 240 | 3% |
| 158714 | 17.7* × 20.9* | 1570 | 1508 | 96% | Bone_1 | 8.2 | 5.2 | 29 | 62 | 4% |
| 158715 | 18.0* × 24.0* | 2481 | 2443 | 98% | Bone_1 | | | 71 | 39 | 2% |
| 158716 | 20.5* × 14.9* | 1245 | 1197 | 96% | Bone_1 | 10.2 | 9.0 | 70 | 48 | 4% |
| 158717 | 18.7* × 19.6* | 1424 | 1071 | 75% | Bone 1 | 18.9 | 12.7 | 14 | 353 | 25% |
| | | Total | 76851 | | | | | Total | 3773 | 5% |

DOI: https://doi.org/10.7554/eLife.34773.008

utilization of bones and are known to consume the entire carcass (freshly killed or scavenged), leaving no bones behind. Bones from hyena kills are left uneaten only during calving seasons of wildebeest, when food (calves) is superabundant, but even those unconsumed bones are eventually eaten after the calving season (*Kruuk, 1972*). Striped and brown hyenas, on the other hand, appear to process bone to a lesser extent based on bone accumulation assemblages (*Wagner, 2006*), and this is reflected in their dental microwear texture (*DeSantis et al., 2017*).

Although carrion feeding and bone consumption are possibly closely associated as increasingly open habitats made carcasses more visible, especially by visual cues from avian scavengers (*Creel, 2001*), bone consumption itself is not always related to scavenging. Competitive social feeding among social predators may be a better predictor of bone eating (*Figure 8*). The earliest bone-crushing dental adaptation in hyaenids, such as *Percrocuta*, appeared in the middle Miocene of Eurasia (*Qiu et al., 1988*; *Ginsburg, 1999*), about 15 million years ago. Before then, this niche was largely occupied by distantly related non-carnivoran carnivorous mammals, such as hyaenodonts, oxyaenodonts, and entelodonts. Canids, however, had actually developed similar adaptations much earlier, such as in the hesperocyonine *Enhydrocyon* in the late Oligocene (more than 28 Ma) of North America (*Wang, 1994*). During the Mio-Pliocene, borophagines had evolved at least two bone-crushing lineages in subtribes Aelurodontina and Borophagina (*Wang et al., 1999*). The advanced genera in these two clades—*Aelurodon* in Aelurodontina, *Epicyon* and *Borophagus* in

**Table 2.** Postcranial specimens used to approximate prey body size based on dimensions of the coprolite rib fragment.

| Family | Genus | Species | Specimen number |
|---|---|---|---|
| Antilocapridae | Antilocapra | americana | LACM 30482 |
| Tayassuidae | Tayassu | pecari | LACM 86904 |
| Camelidae | Lama | guanacoe | LACM 31328 |
| Camelidae | Vicugna | vicugna | LACM 54706 |
| Cervidae | Cervus | axis | LACM 529 |
| Cervidae | Cervus | dama | LACM 30452 |
| Cervidae | Cervus | dama | LACM 30876 |
| Cervidae | Cervus | eldi | LACM 86095 |
| Cervidae | Cervus | nippon | LACM 31069 |
| Cervidae | Cervus | porcinus | LACM 85966 |
| Cervidae | Cervus | timorensis | LACM 86012 |
| Cervidae | Odocoileus | hemionus | LACM 307 |
| Cervidae | Odocoileus | hemionus | LACM 30903 |
| Cervidae | Odocoileus | virginianus | LACM 52442 |

DOI: https://doi.org/10.7554/eLife.34773.011

Borophagina—had independently acquired robust premolars for crushing hard prey items, although their loci of bone-crushing premolars are different (*Figure 7*).

As the oldest living family of carnivorans, canids arose in the late Eocene more than 36 million years ago (*Wang, 1994*; *Wang et al., 2008*). The divergence time between canids and hyaenids should be even earlier than that, tracing back to the initial split of caniforms and feliforms (*Spaulding and Flynn, 2012*). Borophagine canids evolved rather early in canid history in the early Oligocene, about 32 Ma. Hyaenids, by contrast, originated quite late, probably in the early Miocene of Europe, about 21–22 Ma (*Werdelin and Solounias, 1991*; *Ginsburg, 1999*). Despite this late start, hyaenids have evolved the most advanced dentition for crushing bones, seemingly related to their basic feliform dental plan of highly reducing the grinding part of the dentition (M1 and m1 talonid-m2), allowing room for enlargement of their premolars. Canids, on the other hand, are constrained by their less specialized dental plan of retaining a substantial grinding upper and lower molar battery, with less room for premolar enlargement. Therefore, despite a much earlier start, bone-crushing in canids has never advanced to the level of specialization as hyaenids.

Among the living hyaenids, the spotted hyenas have been observed to be the most capable bone-eaters compared to striped hyenas and probably brown hyenas (*Leakey et al., 1999*). However, beyond studies showing differential bone modification at the den sites of different species of hyena, there is not a significant body of research on how striped and brown hyenas hunt differently and how behavioral differences influence their dietary preference as understood from scats (*Watts and Holekamp, 2007*). Given the *Borophagus* coprolite sample and bone-cracking functional morphology (*Figure 6*), the preponderance of evidence points to the striped or brown hyenas as suitable analogs to *Borophagus* in the masticatory and gastrointestinal systems processing bone less thoroughly relative to spotted hyenas. The following discussion regarding the possible ecological role of *Borophagus* should be considered with this difference in mind.

## Bone digestion

The gastrointestinal system of hyenas has apparently evolved to handle large quantities of bones. Hyaenid feces, particularly those of the spotted hyena (*Crocuta crocuta*), are known to contain highly digested calcium phosphates in the form of white powders and bone residues (*Figure 1B*) (*Estes, 1991*). To a lesser extent, the scat of striped hyena (*Hyaena hyaena*) is also white or light gray (*Macdonald, 1978*; *Hulsman et al., 2010*). These white powders consist of calcium and phosphate salts, $Ca_3(PO_4)_2 \cdot 1.5Ca(OH)_2$, similar to hydroxyapatite, the main inorganic component in bones (*Kruuk, 1972*). Assuming that the common ancestor of *Crocuta* and *Hyaena* acquired the

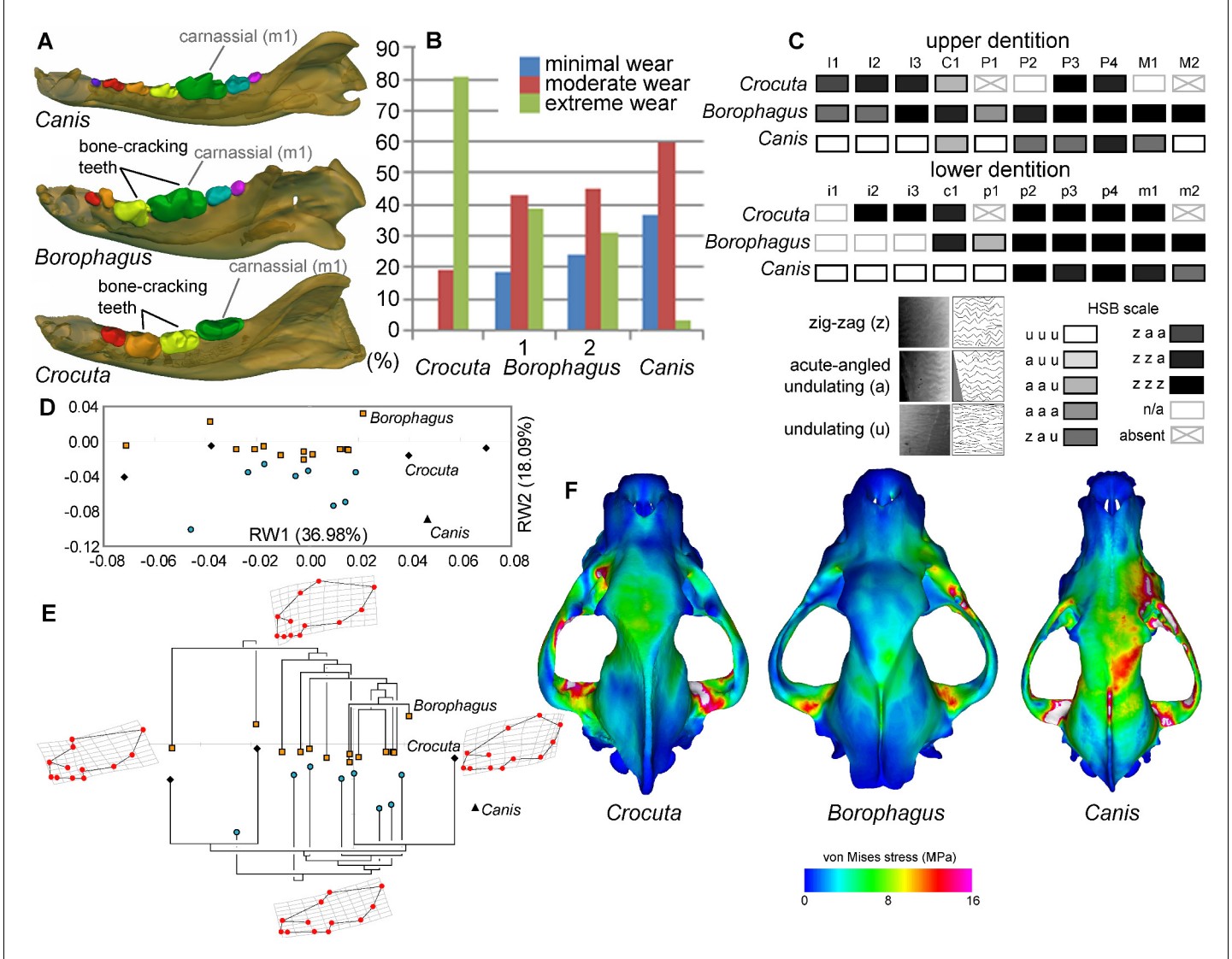

**Figure 6.** Comparison of craniodental functional morphology in *Canis*, *Borophagus*, and *Crocuta*. (A) Lower dentition homology and positions of functionally analogous bone-cracking teeth. Jaws are scaled to the same length. (B) Macrowear data from lower first molar samples of *Crocuta crocuta* (Sub-Saharan Africa) (data from [*DeSantis et al., 2017*]), *Borophagus parvus* (new data based on AMNH specimens from (1) Quibiris Formation, Arizona and (2) Big Sandy Formation, Arizona), and *Canis lupus* (new data based on AMNH specimens from Alberta, Canada). (C) Hunter-Schreger Band (HSB) enamel microstructure patterns in the upper and lower dentitions of the three carnivorans; darker shades indicate higher degree of zig-zag HSB specialization (modified from *Figure 2* from [*Tseng, 2011*]). (D) Morphospace of relative warp (RW) axes from a geometric morphometric analysis of fossil (shaded symbols) and extant (black symbols) canid and hyaenid cranial shape, and (E) Phylogenetic relationships of borophagine canids (top) and hyaenids (bottom) plotted onto morphometric data, with *Canis* indicated by black triangle. Both (D) and (E) are modified from *Figure 5* from *Tseng and Wang (2011)*. (F) von Mises stress distributions in the crania during right fourth premolar bite simulations using 3-D finite element analysis, with warmer colors indicating higher stress. Crania are scaled to the same length (modified from *Figure 7* from [*Tseng, 2013*]).
DOI: https://doi.org/10.7554/eLife.34773.012

bone-dissolving gastrointestinal system, such a trait must have existed more than 8.6 Ma if the molecular divergence time of these two genera is considered (*Koepfli et al., 2006*).

However, despite inferences that the spotted hyena has a highly acidic environment within its gastrointestinal tract, no published measurement is available (*Beasley et al., 2015*). Extant spotted hyenas are also known to regurgitate indigestible contents, such as skin and hair (*Kruuk, 1972*; *Silvestre et al., 2000*). Living domestic dogs have a gastric pH of 1.08–2.07 (*Sagawa et al., 2009*); this is comparable to scavengers with highly acidic stomachs for protection against foreign microbes, such as the turkey vulture (an obligate scavenger; 1.3 ± 0.08) and red-tailed hawk (a facultative

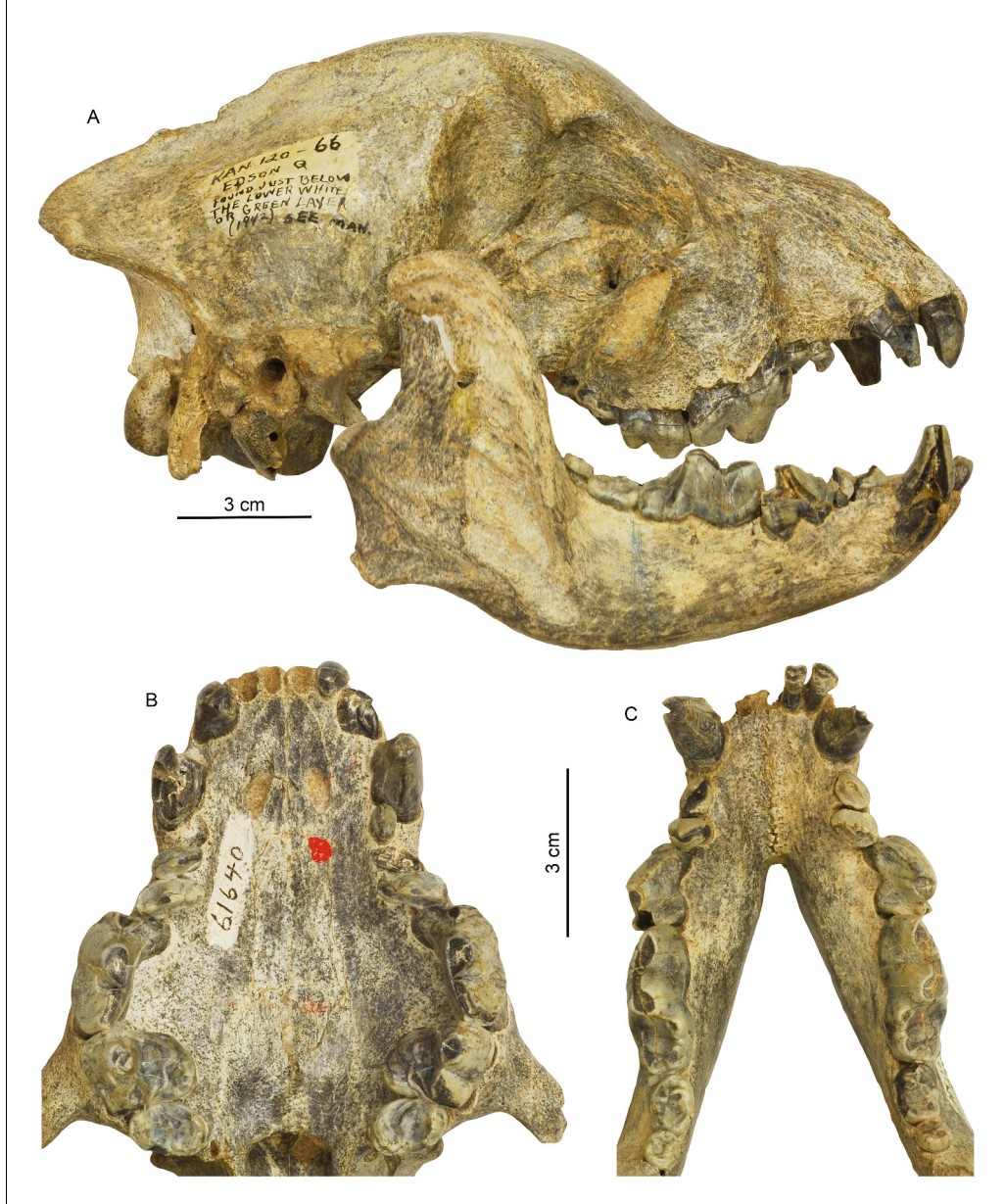

**Figure 7.** Cranial and dental morphology of *Borophagus secundus* (F:AM 61640 from Edson Quarry, Marshall Ranch, Sherman County, Kansas, late Hemphillian). A suite of features is commonly associated with bone-crushing adaptations, such as a highly vaulted forehead, shortened rostrum and associated imbrication of premolars, thickened lower jaws, broadened palate, laterally flared lower cheek teeth, differentially enlarged P4 relative to P3 and p4 relative to p3, and anterior premolars (P1-3 and p1-3) reduced to small pegs that are no longer functioning in occlusion. (**A**) right lateral view of skull and mandible; (**B**) occlusal view of upper teeth; and (**C**) occlusal view of lower teeth.

DOI: https://doi.org/10.7554/eLife.34773.013

scavenger; 1.8 ± 0.27) (***Beasley et al., 2015***). Thus, this hyperacidity in dogs is mainly attributed to scavenging. It is not clear if a linear relationship exists between stomach pH value and the amount of bone residual in scats. Without detailed studies of the digestive process in extant hyenas, it is unknown whether a combination of chemical and mechanical differences in the digestive system is responsible for differences in bone residual size observed between *Borophagus* and living wolves, on one hand, and spotted hyenas on the other. However, examination of the stomach contents of

striped hyenas indicates that they can digest some bones to similar degrees as spotted hyenas (*Kingdon, 1977*).

## Coprolite records

Despite the high concentration of carbonates, modern spotted hyena scat is easily softened and dissolved in the rainy season (*Kruuk, 1972*), and it is not surprising that hyaenid coprolites are rarely preserved in the fossil record. When they are, those from cave hyenas (*Crocuta crocuta spelaea*) are the most common (*Diedrich, 2012*; *Fourvel et al., 2015*; *Sanz et al., 2016*). Highly concentrated

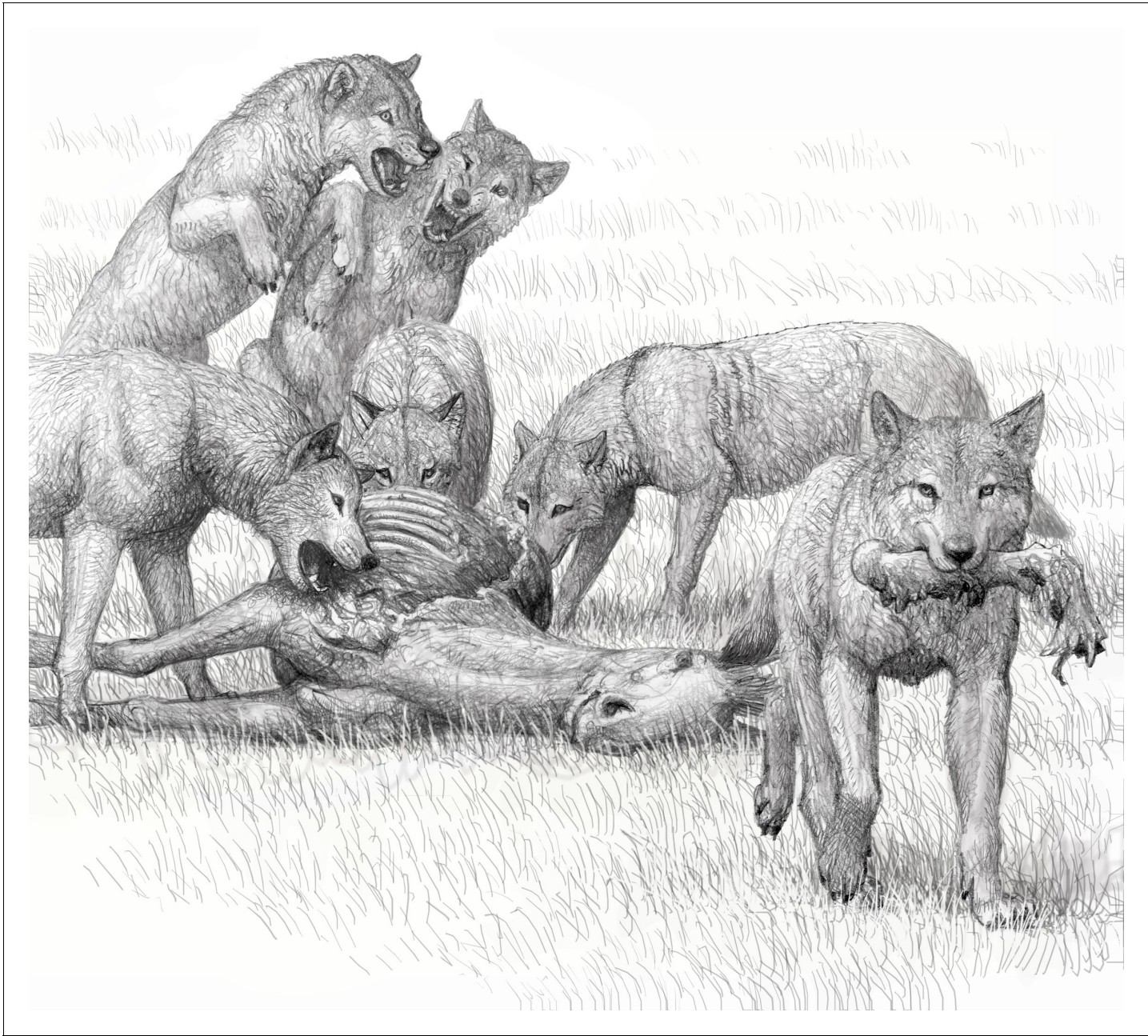

**Figure 8.** Artist conception of feeding by a pack of bone-crushing dogs of the species *Borophagus secundus*, sister taxon of *Borophagus parvus*, by Mauricio Antón. Competitive group feeding does not permit leisurely picking and choosing of meat for quiet consumption and may have been a driving force for complete utilization of carcasses. Adapted from *Wang et al., 2008*: figure 5.4 and with permission for reproduction by Mauricio Antón.
DOI: https://doi.org/10.7554/eLife.34773.014

and trampled feces can result in 'white phosphatic beds', such as in Pleistocene caves in Europe with known cave hyena activities (*Diedrich, 2012*).

In North America, records for canid coprolites are similarly scarce. At the Pipestone Springs Main Pocket site (late Eocene Renova Formation, Jefferson County, Montana), small coprolites have been attributed to *Hesperocyon* (*Lofgren et al., 2015*). Bone fragments inside the coprolites belong mostly to small vertebrates, including marsupials, lizards, lagomorphs, and squirrel-sized rodents, suggesting a diet of mostly small prey. Teeth of *Hesperocyon* have also been reported to occur in some Oligocene coprolites in the Brule Formation from the Big Badlands of South Dakota (*Parris and Holman, 1978*).

## Social hunting

Modern spotted hyenas and wolves are social hunters, and meals are shared by the clans and packs, respectively. Spotted hyenas consume the entire skeleton, bones included, usually in one feeding session (*Kruuk, 1972*). In contrast, wolves are often unable to crack large limb bones, such as those of European bison, and leave substantial parts of the skeleton intact; however, skeletons of smaller prey, such as red deer, suffer far more damage and fewer bones are left uneaten (*Fosse et al., 2012*). In this regard, the bone-processing abilities of wolves are closer to those of brown and striped hyenas than either group is to spotted hyenas. Brown and striped hyenas are solitary foragers and hunters in most observations, although they do have social structures associated with bone accumulations at dens (*Watts and Holekamp, 2007*). Bone-cracking borophagines, such as *Borophagus*, are equipped with far more robust teeth and sturdy jaws than those of extant grey wolves (*Balisi et al., 2018*), although as a clade they did not reach the degree of morphological specialization observed in hyaenids (*Van Valkenburgh et al., 2003*). It is thus reasonable to assume that *Borophagus* is capable of cracking larger bones than living wolves do, possibly comparable to hyaenids. Whether or not *Borophagus* would systematically consume an entire skeleton is still a matter of speculation, but this is likely to depend on the competitiveness of their group feeding.

*Van Valkenburgh et al., 2003* considered large borophagine canids—such as *Epicyon saevus*, *E. haydeni*, *Borophagus secundus*, *Aelurodon ferox*, and *A. taxoides* (*B. parvus* was not included in their study)—to be hunters due to their craniodental morphometrics and abundance in the fossil record, as well as energetic considerations. In contrast to felids that commonly develop a sharp retractile claw as an effective weapon for prey capture (*Gonyea and Ashworth, 1975*), canids never developed a retractile claw (with the possible exception of their arboreal ancestors; [*Wang, 1993*]). *Vanvalkenburgh and Hertel (1993)* and *Van Valkenburgh et al., 2003* thus argued that these large borophagines were likely social hunters in order to overcome their inability to capture large prey by a single individual. Furthermore, *Carbone et al. (1999)* demonstrated an empirical relationship between the body size of carnivorans and their prey size: extant predators of 21.5–25 kg or greater in body mass tend to prey on animals of their own body mass or greater, possibly due to energetic considerations. Our estimate of body mass for Mehrten *B. parvus* is 18.9 ± 1.6 kg based on lengths of the first lower molar or 24.3 ± 3.7 kg based on limb bone circumference and cortical area (see Materials and methods). The latter is generally considered to be more accurate because long bones, as direct weight-bearers, are proportional to body size (e.g., [*Anyonge, 1993*]). Mehrten *B. parvus* thus is comparable in body size to the modern maned wolf *Chrysocyon brachyurus* (23 kg) and African wild dog *Lycaon pictus* (24 kg).

Due to some dental and postcranial parallels between borophagines and modern hyenas, the 'hyaenoid dogs,' as borophagines were earlier known (*VanderHoof and Gregory, 1940*; *Simpson, 1945*), were frequently dismissed as mere scavengers (*Munthe, 1989*) and as such were not presumed to have been able to directly drive the evolution of their prey. Such misconceptions, however, are as much a popular myth about hyenas as a reflection of the fossil dogs. Up to 80% of food consumed by the modern spotted hyena is obtained by their own hunting efforts (*Kruuk, 1972*), in contrast to brown and striped hyenas that are primarily omnivorous scavengers of large prey with less than 5% of food consumed from fresh kills (*Macdonald, 1978*; *Rieger, 1981*; *Mills, 1982*). As active hunters not dependent on the availability of carrion, spotted hyenas typically have a far greater population density and wider distribution than their scavenging relatives. Some large borophagines, such as *Borophagus secundus*, have a continent-wide distribution and abundant fossil record that strongly suggest that they, too, were hunters (*Wang et al., 1999*; *Wang et al., 2008*). In contrast, brown and striped hyenas maintain both smaller species geographic ranges and lower

population densities, both of which are likely associated with their solitary hunting of prey smaller than the preferred prey of spotted hyenas (*Wagner, 2006*). From the new coprolite evidence alone, it is unclear whether *B. parvus* from the Mehrten crossed the size threshold and became an obligate predator of large prey. Our rough body size estimates based on the largest rib fragment inside one of the coprolites (LACM 158708) suggest that the Turlock Lake *Borophagus* probably preyed on ungulates equivalent in size to a modern mule deer *Odocoileus hemionus* (45 to 150 kg), vicuña *Vicugna vicugna* (35 to 65 kg), and guanaco *Lama guanicoe* (90 to 140 kg): animals substantially larger than their own size (see Materials and methods). However, remains of similarly large prey are known from spotted, striped, and brown hyena scats and could represent either scavenged (more likely for striped and brown hyenas) or actively hunted (more likely for spotted hyena) sources (*Kruuk, 1972*; *Wagner, 2006*). Combined with other evidence presented in the preceding paragraphs, the presence of large prey is consistent with—although does not exclusively support—*Borophagus* as social hunters of large mammalian prey.

## Consideration of the ecological role of *Borophagus*

Morphologically hyena-like borophagine canids evolved in and were restricted to North America during their entire fossil record. Around the time of *Borophagus*' extinction towards the end of the Pliocene, and marking the end of hyena-like canid species in North America, a single lineage of hyaenids dispersed to North America (*Berta, 1981*; *Tseng et al., 2013*). One (potentially two) species of the hyaenid *Chasmaporthetes*, like spotted hyenas in their craniodental biomechanical capability (*Tseng et al., 2011*) but with much more cursorially adapted postcranial skeletons (*Berta, 1981*), left a widespread but rare fossil record. Rare fossils of *Chasmaporthetes* from Arizona, Florida, and the Pacific coast of Mexico from otherwise productive localities suggest that either preservational environments were significantly different between *Borophagus* and *Chasmaporthetes* localities, or *Chasmaporthetes* were much less abundant in population density at those localities. Regardless of the reasons for the apparent rarity of the North American hyaenids compared to *Borophagus*, the bone-cracking ecomorphology went extinct in North America no later than the end-Pleistocene megafaunal extinctions. Although there is evidence that another canid, the dire wolf *Canis dirus*, had some degrees of morphological adaptation for consuming hard foods such as bone (*Figueirido et al., 2015*), the selective pressure for such dietary habits may have been short-lived and sensitive to local environmental conditions rather than a long-term macroevolutionary trend (*Van Valkenburgh and Koepfli, 1993*; *DeSantis et al., 2015*).

The distinctive morphological traits associated with the bone-cracking ecomorphology (robust and bulbous premolars, deepened zygomatic arches, arched frontal region, and expanded frontal sinus) are either poorly developed or absent in extant carnivorans (coyotes, foxes, cougars) found today in the geographic regions previously occupied by *Borophagus* (*Werdelin, 1989*; *Tseng and Wang, 2010*; *Tseng and Wang, 2011*). This difference suggests that there is no ecological morphological equivalent of *Borophagus* in modern-day North American food webs. Therefore, the new data and re-interpretation of the functional morphology of *Borophagus* support the inference that their extinction marked the end of a widespread bone-cracking ecomorphology in North America. Combined with the potentially significant role of megafaunal (as opposed to microbial) decomposers such as extant spotted hyenas in influencing or accelerating nutrient cycling pathways and rates by bypassing invertebrate and microbe decomposers in the detrital food web (*Wilson and Wolkovich, 2011*), the extinction of *Borophagus* may have had a much more significant impact on food web dynamics than previously recognized.

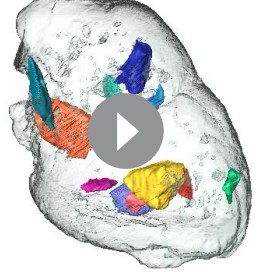

**Video 1.** LACM 158707 movie: A video of microCT scan of LACM 158707 with variously colored bones digitally segmented within the coprolite. Video in Avizo Lite 9.2 by Stuart C. White.
DOI: https://doi.org/10.7554/eLife.34773.009

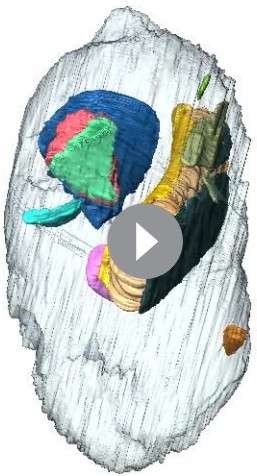

**Video 2.** LACM 158708 movie: A video of microCT scan of LACM 158708 with variously colored bones digitally segmented within the coprolite. Video in Avizo Lite 9.2 by Stuart C. White.
DOI: https://doi.org/10.7554/eLife.34773.010

The above morphological and behavioral comparisons suggest that, regardless of whether *Borophagus* was ecologically equivalent to the top predator spotted hyena or to the small-prey-hunting and large-prey-scavenging brown or striped hyena, such a bone-cracking ecological niche is no longer present in modern-day North American ecosystems. Furthermore, evidence suggests that this change in ecological community composition is a relatively recent phenomenon. (1) Frequent bone consumption in *Borophagus* is supported by both craniodental structure and biomechanics and now (in this study) also by coprolite evidence, suggesting that *Borophagus* may have influenced energy flow in North American food webs similar to what vultures and hyenas may do in Africa today (*DeVault et al., 2003*; *Wilson and Wolkovich, 2011*). (2) Bone digestion in *Borophagus*, as evinced by composition of the coprolites, is less similar to that in extant spotted hyenas and more similar to that in extant wolves and, to some degree, in brown and striped hyenas, suggesting that—top predator or not—*Borophagus* is similar to spotted hyenas in craniodental morphology more than in gastrointestinal physiology, representing a unique combination of traits. (3) The coprolite record of other canids and hyaenids shows that *Borophagus* evolved to consume more bone than earlier canids but did not reach the degree of bone digestion evinced by fossil or living hyaenids. (4) The presence of bone fragments of large mammalian prey is consistent with the interpretation of *Borophagus* as hunters of large prey, like extant wolves and spotted hyenas, but does not preclude a large-prey-scavenging interpretation more similar to the ecological role observed in extant brown and striped hyenas. (5) *Borophagus* fossil sites from the Miocene and Pliocene Epochs cover the area that is, today, nearly the entire continental U.S. into northern Mexico, overlapping with current ranges of predatory canids such as coyotes and foxes and felids such as cougars; these living species are all top predators with little or none of the bone-cracking craniodental morphological characteristics observed in *Borophagus*.

Given these findings, an important future research direction is to examine whether the pre-Ice Age extinction of the hyena-like, bone-eating scavenger represented by *Borophagus* had a fundamental effect on the evolution of food web dynamics (via energy flow modification) during the Ice Age. *Borophagus* was not replaced with a similar ecological morphology on the temporal cusp of the establishment of modern day North American ecosystems. Understanding the impact of such permanent exclusion of a predator/decomposer would be important to understanding sympatric modern food webs.

## Conclusion

Contents from a new sample of coprolites attributed to *Borophagus parvus* from end-Miocene (5.3–6.4 Ma) sediments in northern California provide firsthand insight into the diet of this North American group of bone-cracking top predators. The broad range of bone fragment sizes discovered inside the coprolites suggests that these predators consumed small vertebrate prey as well as deer-sized mammals. Incomplete digestion of prey bones in the coprolites also suggests that, despite a comparable degree of craniodental adaptation for durophagy, canid bone-crackers still possessed a digestive process different from spotted hyenas—which are able to completely break down bone into powder—and were more similar to striped hyenas in this regard. These findings suggest that these bone-cracking canids were potentially social hunters with a unique mixture of typical canid

features and hyena-like characteristics. The ecological niche occupied by the common and widespread *Borophagus* was not replaced by other carnivorans or other mammals after their Pliocene extinction, potentially indicating a fundamental change in food web dynamics in North America as the Ice Age began.

## Materials and methods

All fossil coprolites and associated vertebrate fossils studied from the Mehrten Formation are housed in the Natural History Museum of Los Angeles County (LACM). Additional fossil collections from the Mehrten Formation housed in the Museum of Paleontology at University of California (UCMP) were also studied. Body size estimates are mostly based on comparisons in the Mammalogy collection of the LACM.

Twelve fossil coprolites were scanned in a Skyscan1172 microCT scanner at a pixel resolution of 26.98 micrometers. The resulting basis images were reconstructed and the resulting dataset imported into Fiji (v. 2.0). In Fiji, the image brightness and contrast were optimized, the images were converted from 16 to 8 bits, and the pixels were binned by a factor of two in all planes. This processed image dataset was then imported into Avizo Lite 9.0.1 for analysis. To aid in detecting bone fragments within the coprolite matrix, each image dataset separately was smoothed by applying a 3D median filter to the images using neighborhood values of 6, 18 and 26 pixels. Cross-sectional images were examined using each of the smoothing levels, as well as the original unsmoothed images. Structures identified as bone were marked using the brush tool with the limited range option to most accurately define the bone/matrix edges. Where there were multiple bone fragments in a coprolite, each was marked separately. Each segmented bone fragment included both the cortical and cancellous portions where present (except in the case of a large rib fragment). The Avizo Materials Statistics module was used to determine the volume of each bone fragment and the surrounding matrix. Avizo measurement tools were used to determine the maximum length and width of each bone fragment as well as the orientation of each fragment's long axis with respect to the long axis of the coprolite.

### Fauna, flora, age relationship, depositional setting, and paleoclimate

*VanderHoof (1933)* was first to report a fossil horse, *Pliohippus tantalus*, from near Oakdale in Stanislaus County, California. Although the Oakdale locality has produced only a few fossils since then, it signaled the potential for discovery of vertebrate fossils in the Mehrten Formation, as well as associated plants (*VanderHoof, 1933*; *Axelrod, 1944*). A partial skull of *Megalonyx mathisi* was described subsequently from Black Rascal Creek in 'Upper Mehrten Formation' (*Hirschfeld and Webb, 1968*). Systematic collecting of fossils in the Turlock Lake area, mostly by one of us (DG), was carried out in as early as the 1950 s. In an unpublished Ph.D. dissertation, *Wagner (1981)* reviewed the geologic setting and laid out a biostratigraphic framework of the Mehrten Formation as related to the vertebrate fossils. More recently, *Sankey et al. (2015)* reinvestigated the Turlock Lake fossil sites and began a process of integrating the Mehrten fossils in a modern geologic context (stratigraphic information archived in LACM and UCMP) (see also [*Sankey and Biewer, 2017*]).

The vertebrate fauna from the Modesto Reservoir Member of Mehrten Formation (as defined in [*Wagner, 1981*]) was poorly disseminated and with adequate descriptions of only a few forms: a new 'saber-toothed' salmonid fish *Smilodonichthys rastrosus* (*Cavender and Miller, 1972*; *Sankey et al., 2016*), a bony fish *Orthodon microlepidotus* (*Casteel and Hutchison, 1973*), an extinct New World badger *Pliotaxidea garberi* (*Wagner, 1976*), two plethodontid salamanders *Aneides lugubris* and *Batrachoseps* sp. (*Clark, 1985*), and most recently, a giant tortoise *Hesperotestudo orthopygia* (*Biewer et al., 2014*; *Biewer et al., 2015*; *Biewer et al., 2016*). *Wang et al., 1999* and *Tedford et al., 2009* listed selected borophagine and canine canids from the Mehrten Formation without description or illustration. A systematic revision of Mehrten canids was completed by *Balisi et al. (2018)* that recognized four species: *Borophagus parvus*, *B. secundus*, *Vulpes stenognathus*, and *Eucyon davisi*. See *Wagner (1981)* for a preliminary faunal list of the rest of the unpublished mammals.

The above four canids are all known in the Hemphillian North American Land Mammal age (*Wang et al., 1999*; *Tedford et al., 2009*). *Borophagus parvus*, however, is the most restrictive both in geographic (southwestern United States) and chronologic (late Hemphillian) ranges, offering the

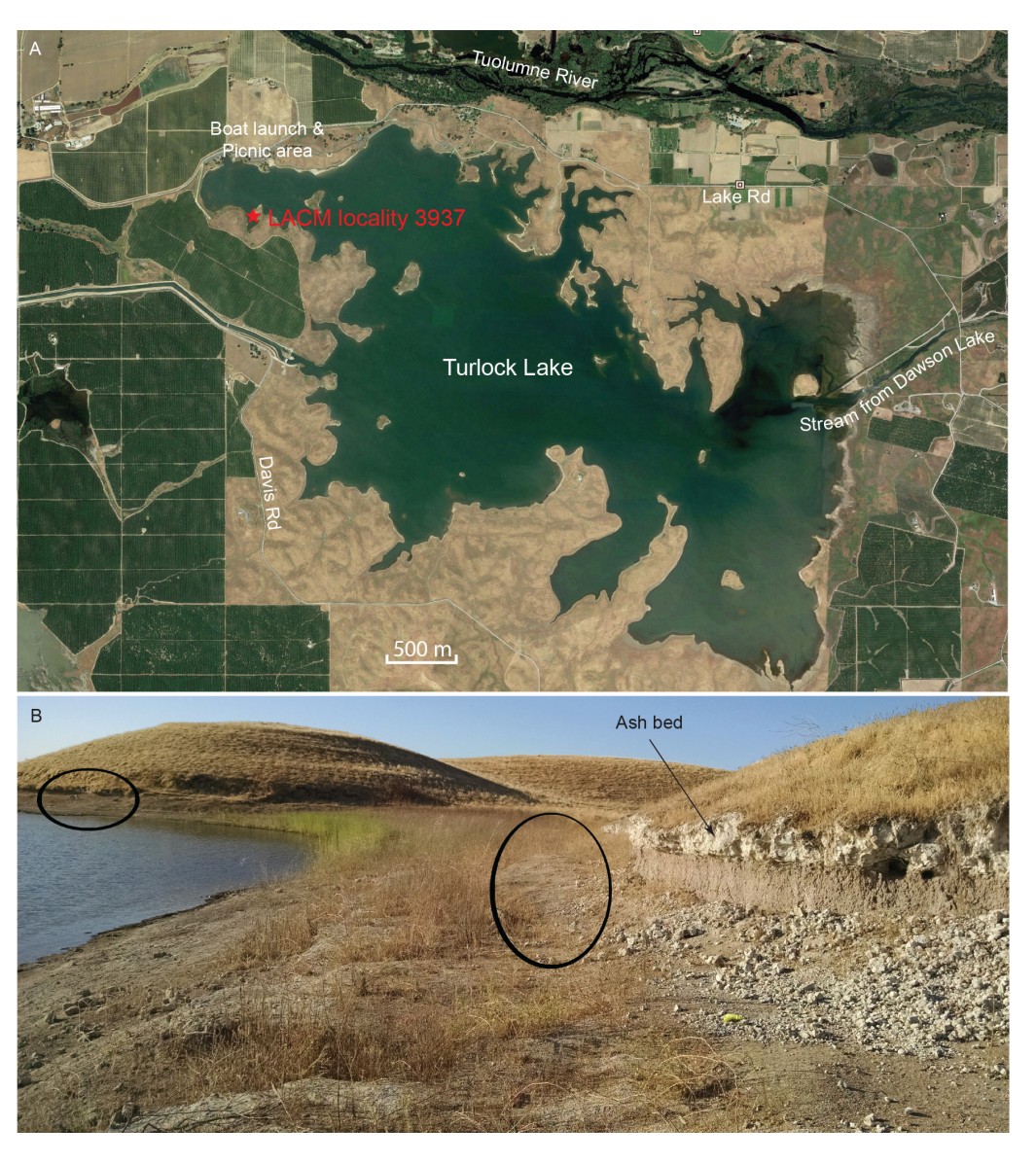

**Figure 9.** Map and photo of coprolite locality. (A) satellite image of Turlock Lake area (37°36–37'N 120°34–36'W) from Google Earth Pro, image date March 31, 2015 (Google Earth Pro (Version 7.1.5.1557), [**Google Inc, 2015**]); red star is approximate position of LACM locality 3937 (=Dennis Garber T-34 locality) and of LACM locality 3935 (=Dennis Garber T-32 locality). (B) LACM locality 3937, looking to the south; black ovals are approximate positions of fossil-producing horizons and that to the left is the location for coprolites; photograph by Jacob Biewer on September 5, 2015.
DOI: https://doi.org/10.7554/eLife.34773.015

best potential for age assessment. *Wang et al. (1999)* commented on the slightly more derived dental characteristic of the Mehrten *B. parvus*, as compared to the topotype materials from the Redington Local Fauna in the lower member of the Quiburis Formation in Pima County, southeastern Arizona, which has been magnetically constrained within Chron 3An.2n (6.436–6.733 Ma) (*Lindsay et al., 1984*; *Hilgen et al., 2012*). If those characters are the result of a chronocline, the Mehrten *B. parvus* may be slightly younger than their Arizona counterpart. *Wagner (1981)* considered the Modesto Reservoir Local Fauna equivalent in age to Pinole Local Fauna in the San Francisco Bay area, which is overlain by a dated tuff (5.3 ± 0.1 Ma) within Pinole Formation and placed in the latest Hemphillian (Hh4) (*Tedford et al., 2004*). If the above comparisons are correct, the Modesto Reservoir Local Fauna should fall in the latest Hemphillian (Hh4), possibly within 5.3–6.4 Ma.

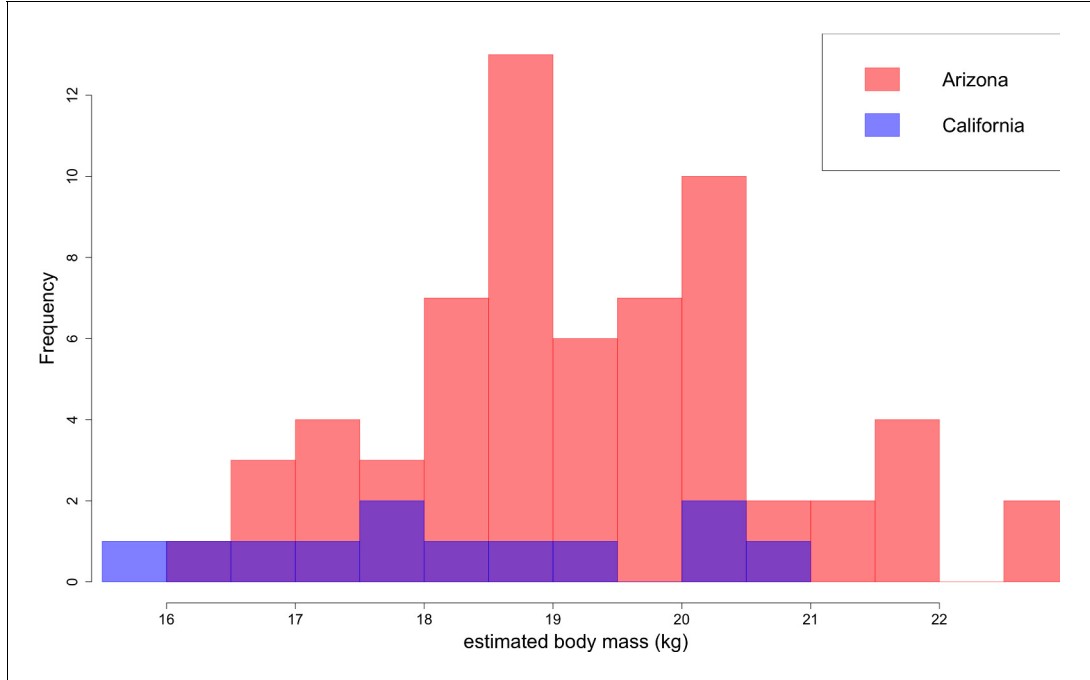

**Figure 10.** Distribution of *B. parvus* body mass estimated from lengths of the lower first molar (carnassial) using the equation from *Van Valkenburgh (1990)*. The Arizona population tends to be larger in body size than the California population, which largely comprises Turlock Lake individuals.
DOI: https://doi.org/10.7554/eLife.34773.016

From two localities, T-14 (LACM locality 3917, Cement Goose Pit Island = UCMP V6878=V90008) and T-20 (LACM locality 3923, Leaf Island), on two small islands in the western part of Turlock Lake (the former shown in *Figure 9A*), *Axelrod (1980)* listed 25 species of fossil plants from what he called Turlock Late Flora consisting of 8 trees (including one conifer), 13 shrubs, 3 herbaceous perennials, and 1 or 2 vines. In particular, aquatic taxa, such as *Cyperus* (flatsedges), *Juncus* (rushes), and *Typha* (cattails), are known to live along the margins of streams, lakes, and ponds, and the fossil plant localities were proposed to be lacustrine deposits 'some distance from the shore' (*Axelrod, 1980*). This flora was characterized as an oak woodland-savanna and a floodplain assemblage, and comparisons to modern vegetation from nearby regions suggested a paleoclimate of slightly cooler (mean annual temperature 15.5°C) and considerably wetter (precipitation 635 mm) than the present-day Turlock Lake area (17.5°C and 335 mm for corresponding measurements). This shift toward a more continental climate in modern day Turlock Lake was suggested to be brought about by the uplift of the Coastal Range and its rain shadow effects during the Pleistocene (*Axelrod, 1980*).

The majority of the coprolites complete enough to be assigned a Dennis Garber field number are produced from a single locality, LACM locality 3937 (=UCMP locality V68134, Dennis Garber T-34 locality), whereas only one coprolite is from LACM locality 3935 (=Dennis Garber T-32 locality) (*Figure 9A*). Both are located on the northwestern corner of Turlock Lake; they are within 300 m of each other and are from approximately the same stratigraphic horizon. T-32 was later subsumed within T-34 as a single locality. Fossil-producing exposures are in a large area forming an elbow shape. At the north end, there are two layers of white volcanic ash sandwiching a brown silty clay (*Figure 9B*). This ash exposure continues to the eastern end of the area, which has a similar lithology to those to the west, although there seems to be a higher ratio of clay to silt and the contact between the ash and clay seems sharper. ([*Retallack, 1997*]:color photo 24) remarked that carnivore coprolites are common in sequences of well-drained soils because of their phosphatic composition and enclosed bones.

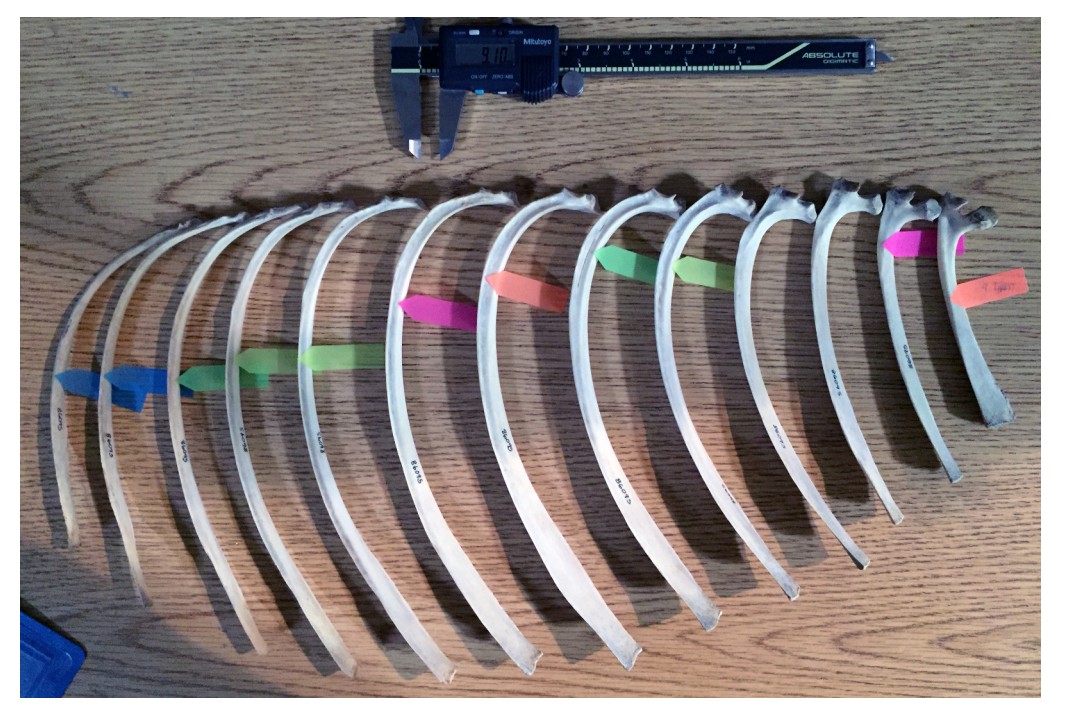

**Figure 11.** Rib measurement methods illustrated on half of a ribcage of Eld's deer (*Cervus eldi*). Anterior ribs are to the right; posterior, to the left. For this set of measurements, the colored tags mark where each rib measures 9.1 mm in anteroposterior width. The corresponding mediolateral thickness at the marked points were then recorded. Ribs without a colored tag were either wider or narrower for much of its length than the two fixed measurements of coprolite width and thickness.

DOI: https://doi.org/10.7554/eLife.34773.017

## Estimation of predator body size

Several regression equations relate skeletal or dental measurements to body mass in extant canids and other carnivorans (*Van Valkenburgh, 1990*; *Anyonge, 1993*; *Anyonge and Roman, 2006*), enabling prediction of the body mass of extinct canids based on measurements of isolated elements. Body mass proxies and their reliability differ slightly, with measures of cross-sectional area of proximal weight-bearing limb bones generating more accurate estimates than dental predictors do. Dental predictors are still useful, however, because teeth tend to be more abundantly preserved than postcrania.

Using the equation of *Van Valkenburgh (1990)*, we generated a distribution of body masses from the lengths of 76 lower first molars (carnassials) of *B. parvus* compiled by *Wang et al. (1999)* and *Balisi et al. (2018)*. We also measured two well preserved *B. parvus* humeri (F:AM 75903-B, F: AM 67955) and one femur (F:AM 63008-A) at the American Museum of Natural History, using the canid equations from *Anyonge (1993)* to calculate body mass from humeral circumference, cortical cross-sectional area, and second moments of area.

Based on lengths of lower first molars, *B. parvus* has a median body mass of 18.9 ± 1.6 kg (*Figure 10*). The Arizona population, with a median mass of 19.2 ± 1.6 kg, tends to be larger in body size than the California population, with a median mass of 18.1 ± 2.0 kg.

Equations using measurements of the humerus and femur, all specimens from the Arizona population, generated higher estimates than dental estimates of both Arizona and California populations. For F:AM 75903-B, a distal humerus, we calculated body mass using an approximation of the circumference (22.8 kg), cortical area (25.829 kg), second moment of area in the anteroposterior plane (20.898 kg), and second moment of area in the mediolateral plane (29.33 kg); these four estimates produced a median measurement of 24.315 ± 3.656 kg. For F:AM 67955, a complete humerus, we obtained a body-size estimate of 32.4 kg using an approximation of the circumference. For F:AM 63008-A, a proximal femur, we estimated 20.2 kg.

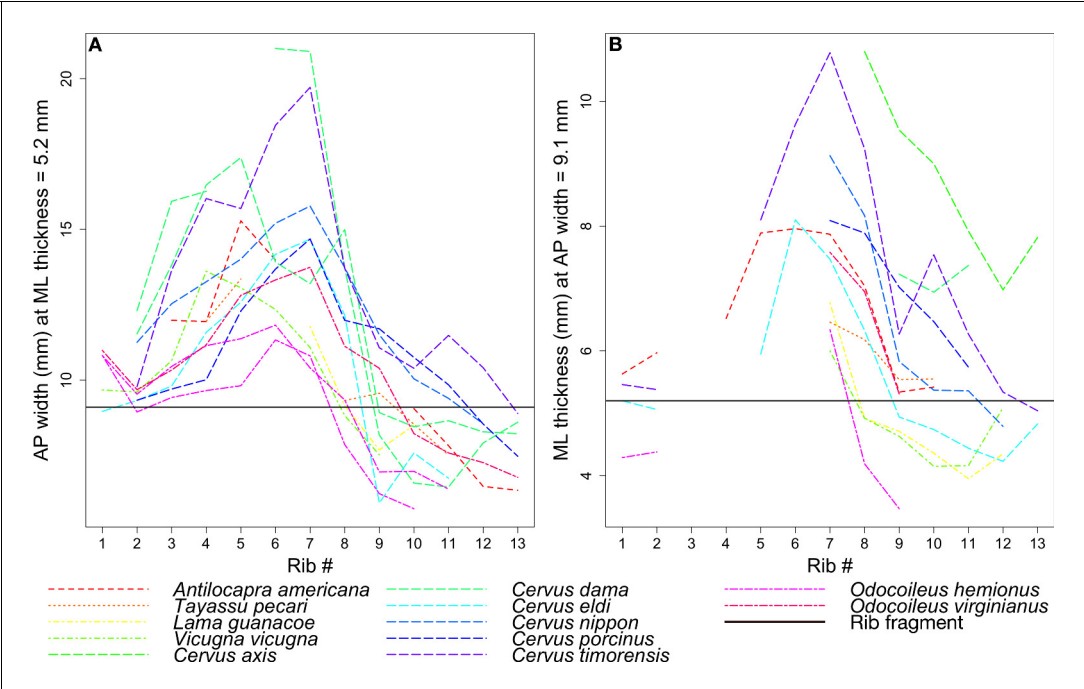

**Figure 12.** Rib measurements per species. The horizontal black line in both plots indicates the corresponding measurement for the coprolite rib fragment. (A) Anteroposterior width of the rib where it has a mediolateral thickness of 5.2 mm, the rib fragment thickness. (B) Mediolateral thickness of the rib where it has an anteroposterior width of 9.1 mm, the rib fragment width.

DOI: https://doi.org/10.7554/eLife.34773.018

These dental and postcranial estimates of body mass place *B. parvus* in the same size class as the dingo *Canis lupus dingo* (20 kg), maned wolf *Chrysocyon brachyurus* (23 kg), African wild dog *Lycaon pictus* (24 kg), red wolf *Canis rufus* (30 kg), and striped hyena *Hyaena hyaena* (35 kg) (**Nowak, 1999**; **Macdonald, 2006**). These extant species (except the omnivorous *Chrysocyon*) are carnivorous to hypercarnivorous.

## Estimation of prey body size

Several species of fossil ungulates have been recorded at Turlock Lake (**Wagner, 1981**), providing a pool of potential prey taxa and a starting point for our analysis. We assembled a comparative rib collection of 14 ungulate individuals belonging to 12 extant species at the LACM, spanning as much as possible the familial diversity preserved at Turlock Lake (**Table 2**). Each specimen comprised a full complement of ribs on at least one side of the body. We attempted to sample the ungulate families recorded by **Wagner (1981)** or, if extinct, the most closely related extant family (e.g. extant Cervidae as proxy for Palaeomerycidae). Extant perissodactyls were not sampled because the perissodactyls at Turlock Lake tend to be either prehistoric equids smaller than modern equids, for which smaller extant artiodactyls could serve as a proxy, or the rhinocerotid *Teleoceras*, which is likely too large to generate the rib fragment preserved in the coprolite.

The coprolite rib fragment has an anteroposterior width of 9.1 mm and mediolateral thickness of 5.2 mm. Using Mitutoyo calipers to the nearest 0.01 mm, we recorded two measurements on each of 13 ribs per species: (1) the mediolateral thickness at the point where it measured 9.1 mm anteroposteriorly, and (2) the anteroposterior width at the point where it measured 5.2 mm mediolaterally (**Figure 11**). The 13 rib measurements per species were visualized using line plots. Species represented by lines intersecting the horizontal line that marked the corresponding coprolite rib measurement were interpreted to be close in size to the prey animal represented by the rib. Because the specimens lacked metadata including body mass, we obtained species body mass estimates from the literature.

**Figure 12** tracks the anteroposterior width or mediolateral thickness of each rib among the extant taxa in comparison to the corresponding measurements in the coprolite rib fragment. Gaps in

the data indicate ribs that were either wider or narrower for much of their length than the two fixed measurements, and so were not measured.

Given its relatively flat morphology, the coprolite rib fragment is unlikely to have come from ribs 1 or 2, which tend to be round in cross-section, despite lines representing these ribs in *Figure 12* intersecting the black line marking the fragment. In general, the ribs examined begin to flatten around rib 3 or 4, and become roughly square around rib 7 to rib 10 before narrowing and rounding again into rib 11 to the end. Therefore, we focused on the species lines that intersect the fragment line only around rib 7 to rib 10.

Given the points of intersection of the species lines with the fragment line, the three ungulate species closest in size to the animal whose rib is preserved in the coprolite are the mule deer *Odocoileus hemionus* (45 to 150 kg), vicuna *Vicugna vicugna* (35 to 65 kg), and guanaco *Lama guanicoe* (90 to 140 kg) (*Nowak, 1999*; *Macdonald, 2006*).

## Acknowledgements

We thank Vanessa Rhue for her assistance in the curation of fossil coprolites at the Natural History Museum of Los Angeles County. Jose Soler and Howell Thomas made casts of fossil canids in this study. Jim Dines and Dave Janiger provided a comparative database for prey body-size estimation with the large-mammal collection at the Natural History Museum of Los Angeles County. We are grateful to Mauricio Antón for permission to reproduce his art works. We also thank Drs Akrivoula Soundia and Sotirios Tetradis of the UCLA School of Dentistry for making the microCT scans. We greatly appreciate reviews by Jessica Thompson, Lars Werdelin, and Katherina Bastl. A research award from the UCLA Department of Ecology and Evolutionary Biology partially funded this study and a Doctoral Dissertation Improvement Grant to MB from the National Science Foundation (DEB-1501931) partially funded this study. The California Department of Parks and Recreation allowed access and permission to work at Turlock Lake. We appreciate funding from the California State University, Stanislaus; the Department of Education (Graduate Assistance in Areas of National Need fellowship); and the Louis Stokes Alliance for Minority Participation (LSAMP).

## Additional information

### Funding

| Funder | Grant reference number | Author |
|---|---|---|
| University of California, Los Angeles | Department of Ecology and Evolutionary Biology | Mairin Balisi |
| National Science Foundation | DEB-1501931 | Mairin Balisi |

The funders had no role in study design, data collection and interpretation, or the decision to submit the work for publication.

### Author contributions

Xiaoming Wang, Conceptualization, Data curation, Formal analysis, Supervision, Investigation, Visualization, Methodology, Writing—original draft, Project administration, Writing—review and editing; Stuart C White, Resources, Data curation, Software, Formal analysis, Validation, Investigation, Visualization, Methodology, Writing—original draft, Writing—review and editing; Mairin Balisi, Resources, Data curation, Formal analysis, Validation, Investigation, Visualization, Methodology, Writing—original draft, Writing—review and editing; Jacob Biewer, Resources, Data curation, Validation, Visualization; Julia Sankey, Resources, Data curation, Supervision, Funding acquisition, Validation, Investigation, Visualization, Writing—original draft; Dennis Garber, Resources, Data curation, Funding acquisition, Validation, Visualization; Z Jack Tseng, Resources, Data curation, Software, Formal analysis, Supervision, Validation, Investigation, Visualization, Methodology, Writing—original draft, Writing—review and editing

## Author ORCIDs
Xiaoming Wang [ID] http://orcid.org/0000-0003-1610-3840
Mairin Balisi [ID] http://orcid.org/0000-0001-6633-1222

## Decision letter and Author response
Decision letter https://doi.org/10.7554/eLife.34773.022
Author response https://doi.org/10.7554/eLife.34773.023

---

## Additional files

### Supplementary files
• Supplementary file 1. 158708 bin=4 Avizo file (plus data folder): Segmentation file for LACM 158708 in Avizo software (ThermoFisher Scientific). Voxel size has been downgraded to 108 micrometers on a side to reduce file size. Segmentation in Avizo Lite 9.2 by Stuart C. White.
DOI: https://doi.org/10.7554/eLife.34773.019

• Transparent reporting form
DOI: https://doi.org/10.7554/eLife.34773.020

### Data availability
All data are published either in the main manuscript or within Supplementary Information. There is no additional data.

---

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
