## [Decision Letter]

Thank you for submitting your article "First bone-cracking dog coprolites provide insight into bone consumption in *Borophagus* and their unique ecological niche" for consideration by *eLife*. Your article has been favorably evaluated by Diethard Tautz (Senior Editor) and three reviewers, one of whom, Jessica Thompson (Reviewer #1), is a member of our Board of Reviewing Editors. The following individuals involved in review of your submission have agreed to reveal their identity: Lars Werdelin (Reviewer #2); Katherina Bastl (Reviewer #3).

The reviewers have discussed the reviews with one another and the Reviewing Editor has drafted this decision to help you prepare a revised submission. Our goal is to provide the essential revision requirements as a single set of instructions, so that you have a clear view of the revisions that are necessary for us to publish your work.

Summary:

This study contributes to our understanding of the structure of the North American carnivore guilds of the Oligocene-Pliocene. It uses data from coprolite size, shape, and bony inclusions to corroborate and extend current work on comparative functional morphology of borophagine dogs and hyenas. The paper offers new independent, empirical evidence for convergent evolution of bone-crushing morphologies/adaptations in multiple carnivore lineages, and examines the role of this adaptation in ecological structure and faunal turnover.

Essential revisions:

This is a well-written and thorough study that makes a novel contribution to understanding convergent evolution of bone-crushing morphologies/adaptations in multiple carnivore lineages. The following essential revisions should be made to enable acceptance.

1) The coprolite data essentially reinforce what was already understood based on bony morphology, so the discussion of the ecological implications should be emphasized. For example, the final sentence is probably the most important insight/conclusion and it would increase the impact of the paper to extend this discussion. This should be done in light of points -4, below.

2) There should be a much more specific discussion of how the bone-crunching adaptation is actually operationalized ecologically, and what implications this would have for adaptations/niches in other lineages when *Borophagus* goes extinct. This should be done through explicit connection to other aspects of anatomy in modern carnivores, especially spotted and striped hyenas, in order to add nuance to the discussion of the ecological positioning of *Borophagus*. This additional discussion does not need to be extensive, but simply more specific and detailed (a reworking of the present discussion), with greater emphasis on explicit comparisons over general descriptions.

3) The focus on wolves and spotted hyenas has turned the Discussion entirely to the topic of "bone-eating top predators" (this begs the question of how a "top predator" is defined and how much bone eating is required for a species to be considered a "bone-eater"), and away from any consideration that *Borophagus* may be more ecologically akin to striped, rather than spotted hyenas. Thus, the section on the "Evolution of the Bone-Eating Top Predator" needs to be either rewritten as a broader discussion of the ecological role of *B. parvus*, or deleted.

4) The authors should seek to find a way to obtain a value for spotted hyena gastric acid if they are to make an argument about diet based on that characteristic. Similarly, a comparison of gut morphology of canids and hyenids and their implications for bone consumption would be useful (specifically, the role of regurgitates versus passing of bone). If a published value (or personal communication from a cited source) truly does not exist for gastric acid, then this section should be de-emphasized because it is otherwise somewhat speculative.

5) Although Figure 7 is appealing, a more instructional Figure 7 would be of the bony morphology of *Borophagus* that provides supporting evidence for the bone-crunching adaptation. The figures would also benefit from more explicit visual comparisons of scats between extant carnivorans at Figure 1, rather than a picture from a website of only hyena scat.

---

## [Author Response]

Essential revisions:This is a well-written and thorough study that makes a novel contribution to understanding convergent evolution of bone-crushing morphologies/adaptations in multiple carnivore lineages. The following essential revisions should be made to enable acceptance.1) The coprolite data essentially reinforce what was already understood based on bony morphology, so the discussion of the ecological implications should be emphasized. For example, the final sentence is probably the most important insight/conclusion and it would increase the impact of the paper to extend this discussion. This should be done in light of points -4, below.

We have added a new section at end of this paper called “Consideration of the ecological role of *Borophagus*” as well as expanding the sections on “Reconstructing the potential ecological role of *Borophagus*” and “Social hunting”. These additions provide substantial new discussion related to ecological implications.

2) There should be a much more specific discussion of how the bone-crunching adaptation is actually operationalized ecologically, and what implications this would have for adaptations/niches in other lineages when Borophagus goes extinct. This should be done through explicit connection to other aspects of anatomy in modern carnivores, especially spotted and striped hyenas, in order to add nuance to the discussion of the ecological positioning of Borophagus. This additional discussion does not need to be extensive, but simply more specific and detailed (a reworking of the present discussion), with greater emphasis on explicit comparisons over general descriptions.

We agree. Our new section (Consideration of the ecological role of *Borophagus*) focuses on the unique combination of borophagine craniodental morphology (strongly trending toward bone-crushing) and their apparently less than thorough digestion of bones. After their extinction before Pleistocene, these distinct morphological traits are either poorly developed or absent among modern North American carnivorans. This suggests that there is no ecological morphological equivalent of *Borophagus* in modern day North American food webs. Bone crushing carnivorans also play a significant role as decomposers, such as extant spotted hyenas, in accelerating nutrient cycling pathways and in bypassing invertebrate and microbe decomposers in the detrital food web.

3) The focus on wolves and spotted hyenas has turned the Discussion entirely to the topic of "bone-eating top predators" (this begs the question of how a "top predator" is defined and how much bone eating is required for a species to be considered a "bone-eater"), and away from any consideration that Borophagus may be more ecologically akin to striped, rather than spotted hyenas. Thus, the section on the "Evolution of the Bone-Eating Top Predator" needs to be either rewritten as a broader discussion of the ecological role of B. parvus, or deleted.

Indeed, the spotted hyena is an extreme example of "bone-eating top predators" in modern carnivore guilds, although bone-crushing adaptations, as shown in craniodental specializations, are not uncommon in the late Cenozoic. We use the spotted hyena as a relatively well-studied case for comparison, but we are mindful that borophagine canids are probably more analogous in its ecomorphology to the brown and stripped hyenas. The latter, however, also have a long history of independent evolution from the canids, with very different starting points in dental batteries and their loci of maximum mechanical bite forces (Werdelin, 1989). We accept the reviewers’ suggestion and added a new paragraph in this section (now called “Reconstructing the potential ecological role of *Borophagus*”) to address their concerns.

4) The authors should seek to find a way to obtain a value for spotted hyena gastric acid if they are to make an argument about diet based on that characteristic. Similarly, a comparison of gut morphology of canids and hyenids and their implications for bone consumption would be useful (specifically, the role of regurgitates versus passing of bone). If a published value (or personal communication from a cited source) truly does not exist for gastric acid, then this section should be de-emphasized because it is otherwise somewhat speculative.

Despite an extensive search (including contacting ecologists and physiologists), we were unable to obtain an acidity value for spotted hyenas stomach from a published source. Therefore, we have tempered our discussion with additional cautionary notes.

5) Although Figure 7 is appealing, a more instructional Figure 7 would be of the bony morphology of Borophagus that provides supporting evidence for the bone-crunching adaptation. The figures would also benefit from more explicit visual comparisons of scats between extant carnivorans at Figure 1, rather than a picture from a website of only hyena scat.

This is a good idea. We added a new Figure 7 featuring a skull, lower jaw, and teeth of *Borophagus secundus*. This figure helps to illustrate typical cranial and dental adaptations in borophagine bone-crushing species. For comparison, we have also added a photo of a wolf scat in Figure 1.